# Imbalanced Semi-Supervised Learning via Label Refinement and Threshold Adjustment

**Zeju Li** *                                                                              *zejuli@fudan.edu.cn*
*College of Biomedical Engineering, Fudan University, Shanghai, China*
*FMRIB Centre, Oxford Centre for Integrative Neuroimaging (OxCIN), University of Oxford, Oxford, UK*

**Ying-Qiu Zheng**                                                                   *ying-qiu.zheng@ndcn.ox.ac.uk*
*FMRIB Centre, Oxford Centre for Integrative Neuroimaging (OxCIN), University of Oxford, Oxford, UK*

**Chen (Cherise) Chen**                                                              *chen.chen2@sheffield.ac.uk*
*School of Computer Science, University of Sheffield, Sheffield, UK*
*Department of Engineering Science, University of Oxford, Oxford, UK*

**Saad Jbabdi**                                                                        *saad.jbabdi@ndcn.ox.ac.uk*
*FMRIB Centre, Oxford Centre for Integrative Neuroimaging (OxCIN), University of Oxford, Oxford, UK*

**Reviewed on OpenReview:** *https://openreview.net/forum?id=HbAMQiyK48*

## Abstract

Semi-supervised learning (SSL) algorithms often struggle to perform well when trained on imbalanced data. In such scenarios, the generated pseudo-labels tend to exhibit a bias toward the majority class, and models relying on these pseudo-labels can further amplify this bias. Existing imbalanced SSL algorithms explore pseudo-labeling strategies based on either pseudo-label refinement (PLR) or threshold adjustment (THA), aiming to mitigate the bias through heuristic-driven designs. However, through a careful statistical analysis, we find that existing strategies are suboptimal: most PLR algorithms are either overly empirical or rely on the unrealistic assumption that models remain well-calibrated throughout training, while most THA algorithms depend on flawed metrics for pseudo-label selection. To address these shortcomings, we first derive the theoretically optimal form of pseudo-labels under class imbalance. This foundation leads to our key contribution: SEmi-supervised learning with pseudo-label optimization based on VALidation data (SEVAL), a unified framework that learns both PLR and THA parameters from a class-balanced subset of training data. By jointly optimizing these components, SEVAL adapts to specific task requirements while ensuring per-class pseudo-label reliability. Our experiments demonstrate that SEVAL outperforms state-of-the-art SSL methods, producing more accurate and effective pseudo-labels across various imbalanced SSL scenarios while remaining compatible with diverse SSL algorithms. The code is publicly available [1].

## 1 Introduction

Semi-supervised learning (SSL) algorithms are trained on datasets that contain labeled and unlabeled samples (Chapelle et al., 2009). SSL improves representation learning and refines decision boundaries without relying on large volumes of labelled data, which are labor-intensive to collect. Many SSL algorithms have

---

*Corresponding author
[1]https://github.com/ZerojumpLine/SEVAL

been introduced, with one of the most prevalent assumptions being consistency (Zhou et al., 2003), which requires the decision boundaries to lie in low density areas. As a means of accomplishing this, pseudo-labels are introduced in the context of SSL (Scudder, 1965), and this concept has been extended to several variants employing diverse pseudo-label generation strategies (Laine & Aila, 2016; Berthelot et al., 2019b;a; Sohn et al., 2020; Wang et al., 2022c). In the pseudo-labelling framework, models trained with labelled data periodically classify the unlabelled samples, and samples that are confidently classified are incorporated into the training set.

The performance of pseudo-label-based SSL algorithms depends on the quality of the pseudo-labels (Chen et al., 2023). In real-world applications, the performance of these SSL algorithms often degrades due to the prevalence of class imbalance in real-world datasets (Liu et al., 2019). When trained with imbalanced training data $\mathcal{X}$, the model $f$ will be biased in inference and tends to predict the majority class (Cao et al., 2019; Li et al., 2020). Consequently, this heightened sensitivity negatively impacts the pseudo-labels produced by SSL algorithms, leading to an ever increasing bias in models trained with these labels.

Therefore, developing improved methods for obtaining high-quality pseudo-labels remains a crucial research direction in imbalanced SSL. Current approaches primarily focus on two strategies: Pseudo-label refinement (PLR) (Lai et al., 2022a; Kim et al., 2020) and threshold adjustment (THA) (Zhang et al., 2021; Wang et al., 2022d). PLR modifies the decision boundary of pseudo-label logits to reduce majority-class bias, while THA optimizes confidence thresholds to maintain an better balance between global diversity and per-class accuracy in the selected pseudo-labels. While numerous SSL algorithms have demonstrated improved performance through enhanced pseudo-label quality, the field lacks a theoretical comparison of these predominantly heuristic-driven approaches. This gap makes it difficult to determine which design best suits specific applications. In this work, we provide new theoretical insights into pseudo-label generation from a statistical perspective, deriving principled strategies for PLR and THA in class-imbalanced scenarios. Surprisingly, our analysis reveals that while existing heuristic solutions partially mitigate bias, they remain fundamentally suboptimal—either lacking theoretical grounding or being compromised by improperly designed metrics for threshold selection.

Our key insight (detailed in Section 3) is that both PLR and THA depend on the distribution of the underlying test data rather than that of the unlabeled training data. Hence, we propose using a small fraction of distinct labeled datasets, as a proxy for unseen test data, to improve the quality of pseudo-labels [2]. Our method is named SEVAL, which is short for SEmi-supervised learning with pseudo-label optimization based on VALidation data. At its core, SEVAL refines the decision boundaries of pseudo-labels using a partition of the training dataset before proceeding with the standard training process. Similarly to AutoML (Zoph & Le, 2016; Ho et al., 2019), SEVAL can adapt to specific tasks by learning from the imbalanced data itself, resulting in a better fit. Moreover, SEVAL learns thresholds that can effectively prioritize the selection of samples from the high-**Precision** class, which we find to be critical but typically overlooked by current model confidence-based dynamic threshold solutions which only focus on **Recall** (Zhang et al., 2021; Guo & Li, 2022).

The contributions of this paper are as follows.

- In the context of pseudo-labeling for imbalanced SSL, we derive the theoretically optimal offsets for PLR and the optimal strategies for THA in Section 3. We also demonstrate the failure cases of existing popular pseudo-labeling strategies.

- We propose learning statistically grounded pseudo-label adjustment offsets in Section 4.1. The derived offsets do not rely on a calibrated model and improve accuracy in both pseudo-labeling and inference.

- We propose learning theoretically grounded thresholds for selecting correctly classified pseudo-labels, using a novel optimization function in Section 4.2. Our strategies outperform existing methods by relaxing the trade-off assumption of **Precision** and **Recall**.

---

[2]Note that in practice, we learn the parameters with a partition of the labeled training dataset before the standard SSL process, thus not requiring any additional data.

- We combine these two techniques into a unified curriculum learning framework based on a training data partition, SEVAL in Section 4.3, and find that it outperforms state-of-the-art pseudo-label based SSL methods such as DARP, Adsh, FlexMatch and DASO under various imbalanced scenarios (c.f. Section 5).

# 2 Related Work

## 2.1 Semi-Supervised learning

Semi-supervised learning has been a longstanding research focus. The majority of approaches have been developed under the assumption of consistency, wherein samples with similar features are expected to exhibit proximity in the label space (Chapelle et al., 2009; Zhou et al., 2003). Compared with graph-based methods (Iscen et al., 2019; Kamnitsas et al., 2018), perturbation-based methods (Xie et al., 2020; Miyato et al., 2018) and generative model-based methods (Li et al., 2017; Gong et al., 2023), using pseudo-labels is a more straightforward and empirically stronger solution for deep neural networks (Van Engelen & Hoos, 2020). It periodically learns from the model itself to encourage entropy minimization (Grandvalet & Bengio, 2004). This process helps to position decision boundaries in low-density areas, resulting in more consistent labeling. Understanding pseudo-label quality holds significance not only for SSL but also for its potential to inspire new applications in other domains (Wu et al., 2025; Sun et al., 2024; Sirbu et al., 2025; Sun et al., 2025).

Deep neural networks are particularly suited for pseudo-label-based approaches due to their strong classification accuracy, enabling them to generate high-quality pseudo-labels (Lee et al., 2013; Van Engelen & Hoos, 2020). Several methods have been explored to generate pseudo-labels with a high level of accuracy (Wang et al., 2022c; Yang et al., 2024). For example, Mean-Teacher (Tarvainen & Valpola, 2017) calculates pseudo-labels using the output of an exponential moving average model along the training iterations; MixMatch (Berthelot et al., 2019b) derives pseudo-labels by averaging the model predictions across various transformed versions of the same sample; FixMatch (Sohn et al., 2020) estimates pseudo-labels of a strongly augmented sample with the model confidence on its weakly augmented version. Many of these approaches falter when faced with class imbalance in the training data, a frequent occurrence in real-world datasets.

## 2.2 Imbalanced Semi-Supervised Learning

Training on imbalanced data biases the model toward majority classes, degrading performance on minority classes (Cao et al., 2019). This bias compounds when using network-generated pseudo-labels: as more samples are pseudo-labeled, the model becomes increasingly skewed toward the majority class. There are three main categories of methods that address this challenge in the literature.

### 2.2.1 Long-tailed learning-based methods

The first group of methods alters the cost function computed using the labeled samples to train a balanced classifier, consequently leading to improved pseudo-labels. Long-tailed learning presents a complex problem in machine learning, wherein models are trained on data with a distribution characterized by a long tail. In such distributions, classes are imbalanced, with the tail classes consistently being underrepresented.

The research on long-tailed recognition (Chawla et al., 2002; Kang et al., 2019; Menon et al., 2020; Tian et al., 2020), which focuses on building balanced classifiers through adjusted cost functions or model structures in a fully supervised learning setting, often serves as inspiration for works in imbalanced SSL. For example, BiS (He et al., 2021) and SimiS (Chen et al., 2022) resample the labelled and pseudo-labelled training datasets to build balanced classifier. ABC (Lee et al., 2021) and Cossl (Fan et al., 2022) decouple the feature learning and classifier learning with a two head model architecture, where a unbiased classifier is learned through weighted cost function with features learned with the original cost function. The key is that they find feature learned with the unnormalized cost function can produce more robust features than that learned from normalized cost functions. Similarly, L2AC (Wang et al., 2022a) further decouples the feature and

classifier learning by building an explicit bias attractor via bi-level optimization. SAW (Lai et al., 2022b) reweights unlabelled samples from different classes based on the learning difficulties. ProCo (Du et al., 2024b) generates balanced pseudo-labels via probabilistic contrastive learning to improve semi-supervised learning (SSL) under class imbalance.

While these strategies can be effectively applied to SSL when assuming correct label generation, their performance remains fundamentally constrained by pseudo-label quality. In contrast, PLR and THA enhance class-wise accuracy by directly modifying pseudo-labels for unlabeled data through distinct mechanisms: PLR reduces class imbalance bias by redirecting selected examples toward adjusted predicted probability distributions; THA balances inter-class trade-offs through selective inclusion criteria for unlabeled examples.

### 2.2.2   Pseudo-label refinement-based methods

PLR-based methods adjust the logits of generated pseudo-labels to reduce the class bias introduced by an imbalanced classifier. The core objective of these approaches is to shift the decision boundary for pseudo-labels by modifying their corresponding logits. A prominent technique within this category is distribution alignment (DA) (Berthelot et al., 2019a), which aims to match the pseudo-label distribution to an estimated class prior. For instance, DARP (Kim et al., 2020) refines pseudo-labels by aligning their distribution with a target distribution, while SaR (Lai et al., 2022a) applies a distribution-alignment mitigation vector to better match the true class proportions.

Alternatively, logit adjustment (LA) (Menon et al., 2020) has been adapted to SSL to counteract class bias by shifting logits according to the logarithm of the ratio of class priors, often derived from labeled data frequencies (Wei & Gan, 2023). LA's inaccuracy for the head class in SSL has motivated several refinements, such as SimPro (Du et al., 2024a), which corrects LA parameters using reliable unlabeled samples, and CoLA (Shao et al., 2026), which incorporates estimated effective numbers and meta-learning into the logit adjustment. Other heuristic approaches have also been explored, including distance-based pseudo-label generation (Oh et al., 2022) and multi-expert frameworks that assign pseudo-labels across different classes (Hou & Jia, 2025), though these methods generally lack theoretical grounding.

Through a statistical analysis in Section 3.2, we demonstrate that DA-based methods are inherently inaccurate because they rely on the distribution of the unlabeled training data, whereas only the true test distribution is theoretically relevant. LA-based methods, while more principled, remain suboptimal as they assume the model is well-calibrated—a condition difficult to satisfy during SSL training. Building on these insights, we derive theoretically motivated logit offsets and propose an optimization strategy for them in Section 4.1.

### 2.2.3   Threshold adjustment-based methods

In SSL, thresholding serves as a critical mechanism designed to filter for the most likely correct pseudo-labels, a technique that has consistently proven effective in enhancing model performance (Sohn et al., 2020). While many early methods focused on refining fixed thresholds through various techniques to improve pseudo-label accuracy, recent studies have demonstrated that moving beyond simple confidence-based criteria can offer significant advantages. For instance, incorporating alternative metrics such as epistemic uncertainty via Monte Carlo dropout (Rizve et al., 2021), energy scores (Yu et al., 2023), or instance-dependent transition matrices (Li et al., 2024) can further benefit the pseudo-labeling process by more effectively filtering out incorrect labels. Furthermore, Dash (Xu et al., 2021) introduces a dynamic element to this process, adjusting thresholds throughout the training duration to gradually incorporate a larger volume of correct samples as the model matures.

Beyond global thresholding strategies, THA-based methods optimize criteria at the class-wise level to address issues like class imbalance. By resampling unlabeled data through these adjusted thresholds, these methods preferentially incorporate samples predicted as minority classes, which acts as a vital form of regularization for the pseudo-labeling process. Adsh (Guo & Li, 2022), for example, utilizes an adaptive thresholding strategy to ensure that a similar proportion of pseudo-labels is selected for each class. Building on the foundations of FixMatch, more advanced frameworks like FlexMatch and FreeMatch (Zhang et al., 2021; Wang et al.,

2022d) select samples based on the model's specific learning progress for each class. This approach allows the system to be more inclusive when the model is not learning well by adjusting expectations accordingly.

Specifically, FlexMatch (Zhang et al., 2021) and related techniques dynamically adjust confidence thresholds by lowering them for classes with lower maximum predicted probabilities—a proxy for **Recall**. This prevents challenging classes from being excluded during training. However, our detailed analysis in Section 3.3 reveals a major flaw in this approach: theoretically, ensuring maximum pseudo-label accuracy requires a focus on **Precision**, not **Recall**. This insight leads us to derive theoretically sound thresholding strategies in Section 4.2.

### 2.2.4 Hybrid solutions

A common trend in SSL is the use of hybrid strategies that jointly optimize loss functions and refine pseudo-labels (Yang et al., 2025). For instance, CReST+ (Wei et al., 2021) combines bootstrap sampling with distribution alignment to mitigate class bias in pseudo-labels. Similarly, DASO (Oh et al., 2022) leverages semantic information to improve pseudo-label quality and aligns balanced prototypes to regularize the feature encoder. CPG (Hou et al., 2025) refines LA parameters and adopts a class-wise data augmentation strategy to enhance minority class representation. ACR (Wei & Gan, 2023) integrates techniques from ABC, FixMatch, and MixMatch, and further employs LA (Menon et al., 2020) to enhance pseudo-labels, achieving strong performance. However, these methods are largely heuristic and often incorporate ad-hoc mechanisms—such as the strong assumptions about unlabeled data distributions in CReST+ and DARP—which can limit their robustness and practical utility in real-world scenarios where such assumptions may not hold.

This proliferation of heuristics stands in contrast to a scarcity of theoretical analysis on what fundamentally improves pseudo-label quality. Without such grounding, it becomes difficult to understand, compare, and advance imbalanced SSL methods systematically. To address this gap, the following section introduces a novel statistical perspective that disentangles the core components of SSL algorithms and establishes a theoretical foundation for existing approaches. Building on this formulation, we present SEVAL: a flexible method that integrates directly into standard SSL pipelines without requiring changes to the learning strategy, data sampling, or extra pseudo-label computations. SEVAL makes no assumptions about label distributions, making it applicable to any dataset imbalance setting. By clearly decoupling the SSL design space in both methodology and evaluation, our work enables fairer comparisons across different solution categories and offers a clearer pathway for future research.

## 3 Limitations of Current Methods

In this section, we begin by summarizing the framework of current pseudo-label based SSL. Next, we break down the design of current imbalanced SSL into two key components: PLR and THA. We then offer insights into these components through theoretical analysis, highlighting previously overlooked aspects.

### 3.1 Pseudo-label-based SSL

We consider the problem of $C$-class semi-supervised classification. Let $X$ be the input space and $Y = \{1, 2, \ldots, C\}$ be the label space. We are given a set of $N$ labelled samples $\mathcal{X} = \{(\boldsymbol{x}_i, y_i)\}_{i=1}^{N}$ and a set of $M$ unlabelled samples $\{\boldsymbol{u}_i\}_{i=1}^{M}$ in order to learn an optimal function or model $f$ that maps the input feature space to the label space $f : X \to \mathbb{R}_{+}^{C}$. $f$ can be viewed as an estimate of the conditional probability distribution $P(Y|X)$. In deep learning, $f$ can be implemented as a network followed by a softmax function $\sigma$ in order to produce the probability score $p_c$ for each class $c$: $p_c = \sigma(\boldsymbol{z})_c = \frac{e^{z_c}}{\sum_{j=1}^{C} e^{z_j}}$ where $\boldsymbol{z}$ is the raw output (in $C$-dim) produced by the network. The predicted class is assigned to the one with the highest probability.

The parameters of $f$ can be optimized by minimizing an empirical risk computed from the labeled dataset using:

$$R_{\mathcal{L},\mathcal{X}}(f) = \frac{1}{N}\sum_{i=1}^{N}\mathcal{L}(y_i, \boldsymbol{p}_i^{\mathcal{X}}), \tag{1}$$

where $\boldsymbol{p}_i^{\mathcal{X}}$ is the model predicted probabilities on the labelled data and $\mathcal{L}$ can be implemented as the most commonly used cross-entropy loss. To further utilize unlabelled data, a common approach is to apply pseudo-labeling to unlabelled data where the estimated label is generated from the network's predicted probability vector $\boldsymbol{q} \in \mathbb{R}^C$ (Lee et al., 2013). As there is no ground truth, we use the current model to produce a pseudo-label probability vector $\boldsymbol{q}_i \in \mathbb{R}^C$ for an unlabelled sample $\boldsymbol{u}_i$ and the pseudo-label $\hat{y}_i$ is determined as $\arg\max_j(q_{ij})$. In this way, we obtain $\hat{\mathcal{U}} = \{(\boldsymbol{u}_i, \hat{y}_i)\}_{i=1}^{M}$, where each $(\boldsymbol{u}_i, \hat{y}_i) \in (X \times Y)$. As the predicted label quality can be very poor especially at the early stage and for some challenging data, it is common to apply a threshold to identify reliable labels for model optimization [3]. The risk function on the unlabelled data can be defined as:

$$\hat{R}_{\mathcal{L},\hat{\mathcal{U}}}(f) = \frac{1}{M}\sum_{i=1}^{M}\mathbb{1}(\max_j(q_{ij}) \geq \tau)\mathcal{L}(\hat{y}_i, \boldsymbol{p}_i^{\mathcal{U}}), \tag{2}$$

where $\mathbb{1}$ is the indicator function and $\tau$ is a predefined threshold that filters out pseudo-labels with low confidence, and $\boldsymbol{p}_i^{\mathcal{U}}$ is the predicted probability for the unlabelled data. Unless otherwise specified, we do not have specific requirements for the network architecture in our setting. The neural network is trained using an equally weighted combination of the risk functions for labeled data $R_{\mathcal{L},\mathcal{X}}(f)$ and $\hat{R}_{\mathcal{L},\hat{\mathcal{U}}}(f)$.

In a class imbalanced setting, we have a challenge: the distribution of samples across the $C$ classes is highly uneven with varying numbers of samples per class $\mathbf{n} : n_1, ..., n_c$. Some classes contain abundant samples (majority classes), while others have very few (minority classes). The class imbalance ratio is defined as $\gamma = \frac{max(\mathbf{n})}{min(\mathbf{n})}$, which in typical imbalanced settings exceeds 10 (reaching 30 in naturally collected datasets (Su & Maji, 2021)). In such case, the pseudo-labels on the unlabelled data can be biased to the majority class, which further amplifies the class imbalance problem. The model tends to predict majority classes over true classes during testing.

In this paper, to alleviate the issue of class imbalance-induced bias, we propose a method which can refine the probability vector used for pseudo-labelling $\boldsymbol{q}$ (c.f. Section 4.1) [4]. We also propose to adjust the threshold to operate on a class-specific basis, i.e. we use a vector $\boldsymbol{\tau} \in \mathbb{R}^C$ of threshold values to achieve accuracy fairness (c.f. Section 4.2). The model can then dynamically select the appropriate thresholds based on its prediction. The primary contribution of this paper is the statistical analysis of optimal strategies for adjusting $\boldsymbol{q}$, as well as the theoretical optimal choice of $\boldsymbol{\tau}$. In the following section, we will bypass the computation of pseudo-label probability $\boldsymbol{q}_i$ and concentrate on our contributions.

### 3.2 Pseudo-Label Refinement (PLR)

For an unlabelled sample $\boldsymbol{u}_i$, we determine its pseudo-label probability $\boldsymbol{q}_i$ based on its corresponding pseudo-label logit $\hat{\boldsymbol{z}}_i^{\mathcal{U}}$. In the process of PLR, we aim to adjust the decision boundaries for $\hat{\boldsymbol{z}}_i^{\mathcal{U}}$ with offset $\boldsymbol{\pi} \in \mathbb{R}^C$ to reduce class biases. Many methods in the literature have been discussed to utilize different $\boldsymbol{\pi}$ from different perspectives, such as DA and LA. However, we find that none of them provides accurate refinement for imbalanced SSL. Here, we aim to shed new light on this problem by analyzing optimal THA strategies from a statistical perspective (Saerens et al., 2002). We assume that the test distribution $\mathcal{T}$ shares identical class conditionals with the training dataset $\mathcal{X}$ (i.e., $P^{\mathcal{X}}(X|Y) = P^{\mathcal{T}}(X|Y)$) and deviates solely in terms of class priors ($P^{\mathcal{X}}(Y) \neq P^{\mathcal{T}}(Y)$), we can assert:

---

[3]We describe the case of hard pseudo-labels for simplicity, but the method generalizes to the case of soft pseudo-labels.

[4]We use different notations $\boldsymbol{q}$ and $\boldsymbol{p}^{\mathcal{U}}$ to denote the probability obtained for pseudo-labelling and model optimization. This is common in existing SSL algorithms, where the two can be obtained in different ways for better model generalization (Laine & Aila, 2016; Sohn et al., 2020; Berthelot et al., 2019b;a). For example, in FixMatch (Sohn et al., 2020), $\boldsymbol{q}_i = f(\mathcal{O}_w(\boldsymbol{u}_i))$ is estimated on a weakly augmented sample of an input image for reliable supervision whereas $\boldsymbol{p}_i^{\mathcal{U}}$ is estimated using a strongly-augmented (i.e. RandAugment (Cubuk et al., 2020)) version $\mathcal{O}_s(\boldsymbol{u}_i)$ as model input for the same instance $i$.

Table 1: Theoretical comparisons of SEVAL and other pseudo-label refinement methods including distribution alignment (DA) (Berthelot et al., 2019a; Wei et al., 2021; Lai et al., 2022a; Kim et al., 2020), logit adjustment (LA) (Wei & Gan, 2023; Menon et al., 2020) and DASO (Oh et al., 2022). $\mathcal{X}$: Labelled training data; $\mathcal{U}$: Unlabelled training data; $\mathcal{T}$: Test data; $\mathcal{V}$: An independent labelled data.

| | Optimal results (c.f. Eq. 1) | DA (Berthelot et al., 2019a) | LA (Menon et al., 2020) | DASO (Oh et al., 2022) | SEVAL (c.f. Section 4.1) |
|---|---|---|---|---|---|
| Estimation of the classifier on re-sampled $\mathcal{U}$, $f^{\mathcal{U}}(X)$ | $\dfrac{f^*(X)P^{\mathcal{T}}(Y)}{P^{\mathcal{X}}(Y)}$ | $\dfrac{f(X)P^{\mathcal{U}}(Y)}{\hat{P}^{\mathcal{U}}(Y)}$ | $\dfrac{f(X)P^{\mathcal{T}}(Y)}{P^{\mathcal{X}}(Y)}$ | Blending similarity based pseudo-label | $\dfrac{f(X)P^{\mathcal{T}}(Y)}{\boldsymbol{\pi}^*}$ |
| Note | In the spirit of statistics. | $\hat{P}^{\mathcal{U}}(Y)$ is the model prediction of $P^{\mathcal{U}}(Y)$. Suboptimal as it ignores $P^{\mathcal{T}}(X,Y)$, especially with class imbalance. | Inaccurate as $f$ is suboptimal and uncalibrated. | Relying on the effectiveness of blending strategies. | Optimizing the decision boundary on $\mathcal{U}$ using $\mathcal{V}$ as a proxy without assuming a specific $f$. |

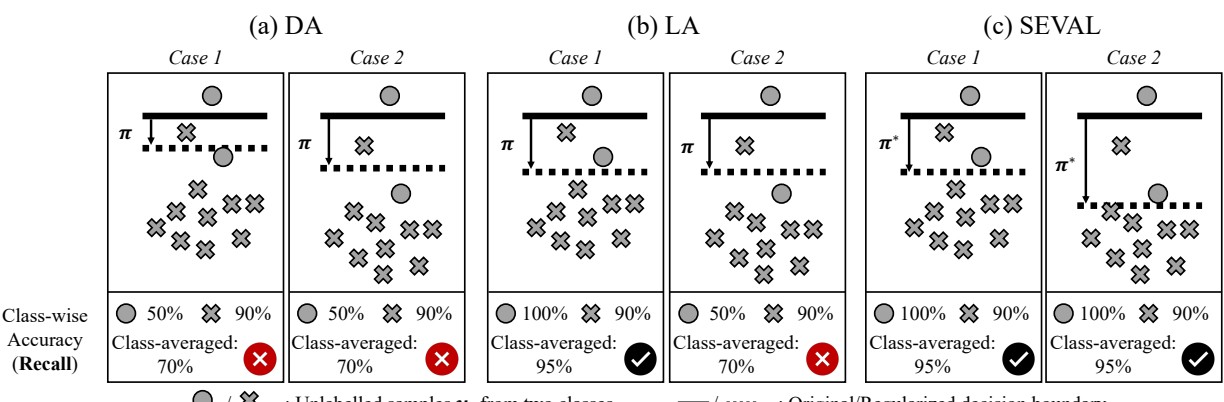

Figure 1: Illustration of limitations in PLR methods for a two-class imbalanced classification task. As in Remark 1, assuming $P^{\mathcal{T}}(Y)$ is uniform, the decision boundary should shift to maximize class-averaged likelihood. Here, we show class-wise accuracy at the bottom, which can be seen as an indictor of the class-averaged likelihood but is more straightforward to compare. (a) DA imposes constraints based on the distribution of $\mathcal{U}$. Even if $\mathcal{U}$ is estimable, the regularized boundary tends to be unfair to minority classes due to class imbalance, as false negatives can have a greater impact on the minority class. (b) LA refines the boundary using training data distributions, but fixed offsets (e.g., $\boldsymbol{\pi}$) is likely to perform poorly as logit distributions shift during training. (c) The proposed scheme directly optimizes the class-averaged likelihood (c.f. Section 4.1) and fits the training process with curriculum learning (c.f. Section 4.3).

**Proposition 1** *Given that a classifier $f^*(X)$ is optimized on $P^{\mathcal{X}}(X,Y)$,*

$$f^{\mathcal{T}}(X) \propto \frac{f^*(X)P^{\mathcal{T}}(Y)}{P^{\mathcal{X}}(Y)}, \tag{3}$$

*is the optimal* Bayes classifier *on $P^{\mathcal{T}}(X,Y)$, where $P^{\mathcal{X}}(X|Y) = P^{\mathcal{T}}(X|Y)$ and $P^{\mathcal{X}}(Y) \neq P^{\mathcal{T}}(Y)$.*

If we have access to the class ratio between underlying test data and unlabeled training data, we can construct a reweighted dataset by adjusting $(X,Y)$ to align the label distribution of the unlabeled training dataset $P^{\mathcal{U}}(Y)$ with that of the target dataset $P^{\mathcal{T}}(Y)$, while preserving the feature-label relationship $P^{\mathcal{U}}(X,Y)$. Using this resampled dataset, we can assert:

**Corollary 1** *The Bayes classifier $f^{\mathcal{T}}(X)$ should be also optimal on the resampled unlabelled data $\dfrac{P^{\mathcal{U}}(X,Y)P^{\mathcal{T}}(Y)}{P^{\mathcal{U}}(Y)}$, where $P^{\mathcal{T}}(X|Y) = P^{\mathcal{U}}(X|Y)$ and $P^{\mathcal{T}}(Y) \neq P^{\mathcal{U}}(Y)$.*

We provide proofs in Appendix Section C.1. This analysis provides insight into the formulation of pseudo-label offsets: *it depends on the label distribution of the test data, $P^{\mathcal{T}}(Y)$, rather than the distribution of unlabeled data, $P^{\mathcal{U}}(Y)$.*

According to Corollary 1, in order to learn the optimal Bayes classifier on $P^{\mathcal{U}}(X,Y)$, we should resample or reweight the unlabelled data based on the class priors ratio of $P^{\mathcal{T}}(Y)/P^{\mathcal{U}}(Y)$. If $P^{\mathcal{T}}(Y)$ follows a uniform distribution [5], the classifier should be trained to achieve optimal performance on $P^{\mathcal{U}}(X,Y)$ across different classes as the term $P^{\mathcal{T}}(Y)/P^{\mathcal{U}}(Y)$ would normalize the optimization functions across classes. Specifically, we claim that:

**Remark 1** *The optimal Bayes classifier on $P^{\mathcal{U}}(X,Y)$ should have maximized class-averaged likelihood, if the $P^{\mathcal{T}}(Y)$ is uniform.*

From this perspective, we summarize current PLR approaches in Table 1. We find that although existing approaches can reduce the class bias of pseudo label by moving the decision boundary based on different criteria, their discrepancy with the optimal results would lead to suboptimal results.

DA (Berthelot et al., 2019a; Wei et al., 2021; Kim et al., 2020) is a commonly employed technique to make balanced prediction for different classes which align the predicted class priors to true class priors of $\mathcal{U}$, making the model being fair (Bridle et al., 1991). Typically, it is only applicable to cases where the labeled and unlabeled datasets have the same label distributions, as it requires the assessment of the true class prior of the unlabeled data. However, this is not necessarily true in real-world applications. More importantly, it only reduces the calibration errors but cannot be optimally fair because it relies on an incorrect optimization function that does not take $P^{\mathcal{T}}$ into account. Fig 1(a) illustrates this in imbalanced SSL. The decision boundary obtained through DA would be suboptimal because enforcing proportional predictions (i.e., requiring two sample predictions for class ◉) increases false negatives without improving true positives.

LA modifies the network prediction from $\arg\max_c(\hat{z}_{ic}^{\mathcal{U}})$ to $\arg\max_c(\hat{z}_{ic}^{\mathcal{U}} - \beta\log\pi_c)$, where $\beta$ is a hyper-parameter and $\boldsymbol{\pi}$ is determined as the empirical class frequency (Menon et al., 2020; Zhou & Liu, 2005; Lazarow et al., 2023). It shares similar design with Eq. 1, however, recall that proposition 1 provides a justification for employing logit thresholding when optimal probabilities $f^*(X)$ are accessible. Although neural networks strive to mimic these probabilities, it is not realistic for LA as neural networks are often uncalibrated and over confident (Guo et al., 2017). We think that LA could have more profound errors when applied to SSL, as the classifier is not optimal and the model is likely to be uncalibrated during the training process (Loh et al., 2022). An obvious failing case is illustrated in Fig. 1(b), where *Case 1* and *Case 2* represent the logit distributions of different training procedures. During the training process, the logits of samples are pushed away from the decision boundary (i.e. *Case 1 $\to$ Case 2*), and the optimal regularized decision boundary should also shift. However, the decision boundary refinement in LA is only related to the class prior of the training and test datasets, and it lacks a mechanism to dynamically track the logit distribution.

The accurate estimation of classifier bias requires the calculation of a conditional confusion matrix, which always requires holdout data (Lipton et al., 2018). Therefore, in this study, we estimate the classifier bias using holdout data and explicitly determine the optimal decision boundary following Remark 1. We hypothesize that the optimal decision boundary should remain consistent across datasets with similar training sample sizes. Accordingly, we develop a curriculum learning approach that: (1) learns optimal parameters by splitting the training dataset, and (2) applies the derived patterns to the complete training dataset. This methodology enables optimal decision boundary derivation during training without requiring additional labeled data. The importance of a curriculum in SSL—where dynamic pseudo-labels are used at

---

[5] For simplicity, here and in the following section, we assume that the test data $P^{\mathcal{T}}(Y)$ is equally distributed across different classes, which is the most common assumption in existing methods. The analysis and methods can be readily extended to other assumptions regarding $P^{\mathcal{T}}(Y)$, as we discuss in Appendix E.

different learning stages—is also highlighted in (Cascante-Bonilla et al., 2021). Further details are provided in Section 4.1.

### 3.3 Threshold Adjustment (THA)

Here, we look into the impact of pseudo-label quality on the SSL. For simplicity, we consider in this section a model $f$ trained on $\hat{\mathcal{U}}$ for *binary classification* using supervised learning loss $\mathcal{L}$. $(\boldsymbol{u}, \hat{y})$ is an arbitrary sample drawn from $\hat{\mathcal{U}}$. In this section we refer $\mathcal{U} = \{(\boldsymbol{u}_i, y_i)\}_{i=1}^M$ to the oracle distribution which contains the inaccessible real label $y$ for $\boldsymbol{u}$. Let $\rho$ be the noise rate of selected pseudo-labels, defined as $\rho = 1 - P_{\hat{\mathcal{U}}}(y = \hat{y})$ with $\rho < 0.5$. The oracle expected risk is defined as $R_{\mathcal{L}, \mathcal{U}}(f) := \mathbb{E}_{(\boldsymbol{u}, y) \sim \mathcal{U}}[\mathcal{L}(f(\boldsymbol{u}), y)]$.

The theorem presented below indicates that, with the number of samples in set $\mathcal{U}$ is fixed as $M$, better model performance is achieved through training with a dataset exhibiting a lower noise rate $\rho$.

**Theorem 1** *Given $\hat{f}$ is the model after optimizing with $\hat{\mathcal{U}}$. Assume the loss function $\mathcal{L}$ is $L$-Lipschitz in all predictions. For any confidence parameter $\delta > 0$, the following holds with probability $\geq 1 - \delta$:*

$$R_{\mathcal{L}, \mathcal{U}}(\hat{f}) \leq \min_{f \in \mathcal{F}} R_{\mathcal{L}, \mathcal{U}}(f) + 4 L_p \mathfrak{R}(\mathcal{F}) + 2\sqrt{\frac{\log(1/\delta)}{2M}}, \tag{4}$$

*where the Rademacher complexity $\mathfrak{R}(\mathcal{F})$ is defined by $\mathfrak{R}(\mathcal{F}) := \mathbb{E}_{\boldsymbol{x}_i, \epsilon_i}\left[\sup_{f \in \mathcal{F}} \frac{1}{M} \sum_{i=1}^M \epsilon_i f(\boldsymbol{u}_i)\right]$ for function class $\mathcal{F}$ and $\epsilon_1, \ldots, \epsilon_M$ are i.i.d. Rademacher variables. $L_p \leq \frac{2L}{1-2\rho}$ is the Lipschitz constant.*

We provide proofs in Appendix Section C.3.

The insight of Theorem 1 is straightforward: given a fixed label size $M$, reducing the noise rate $\rho$ tightens the bound toward its upper limit, yielding a better model. To achieve this, we want to increase the accuracy of the pseudo-label, of which the marginal distribution over $\hat{y}$ can be calculated as:

$$P_{\hat{\mathcal{U}}}(y = \hat{y}) = \sum_{j=1}^C \underbrace{P_{\hat{\mathcal{U}}}(y = j | \hat{y} = j)}_{\textbf{Precision}} \underbrace{P_{\hat{\mathcal{U}}}(\hat{y} = j)}_{Accessible}. \tag{5}$$

This indicates that, if we can have an approximation of **Precision**, we can feasibly tune the pseudo-label accuracy by controlling the sampling strategies based on class-specific thresholds $\boldsymbol{\tau}$. However, it is not possible with

$$P_{\hat{\mathcal{U}}}(y = \hat{y}) = \sum_{j=1}^C \underbrace{P_{\hat{\mathcal{U}}}(\hat{y} = j | y = j)}_{\textbf{Recall}} \underbrace{P_{\hat{\mathcal{U}}}(y = j)}_{Inaccessible}, \tag{6}$$

because in practice we do not have the information of ground truth label for $\boldsymbol{u}_i$. Formally, we claim that:

**Remark 2** *A better thresholding vector $\boldsymbol{\tau}$ for selecting effective pseudo-labels should be derived from class-wise **Precision** rather than class-wise **Recall**.*

However, existing dynamic threshold approaches (Zhang et al., 2021; Wang et al., 2022d; Guo & Li, 2022) derive the threshold for class $c$ based on estimated **Recall**. Specifically, they all rely on the maximum class probability of class $c$, i.e. $P_c'$ [6]. Although sample-wise aggregation of maximum class probabilities estimates overall accuracy (Guo et al., 2017) of test samples (Garg et al., 2022; Li et al., 2022), it is important to note that when evaluated per-class, $P_c'$ corresponds to **Recall** since negative samples are excluded. In other words, when dynamic techniques like FlexMatch prioritize selecting samples from classes with lower maximum class probabilities, they inherently increase sampling frequency for classes that demonstrate lower **Recall**!

---

[6]Formally, it is calculated as $P_c' = \frac{1}{K_c} \sum_{i=1}^K \mathbb{1}_{ic} \max_j p_{ij}^{\mathcal{U}}$, where $\mathbb{1}_{ic} = \mathbb{1}(\arg\max_j(p_{ij}^{\mathcal{Y}}) = c)$ is 1 if the predicted most probable class of sample $i$ is c and 0 otherwise and $K_c = \sum_{i=1}^K \mathbb{1}_{ic}$ counts samples predicted as $c$.

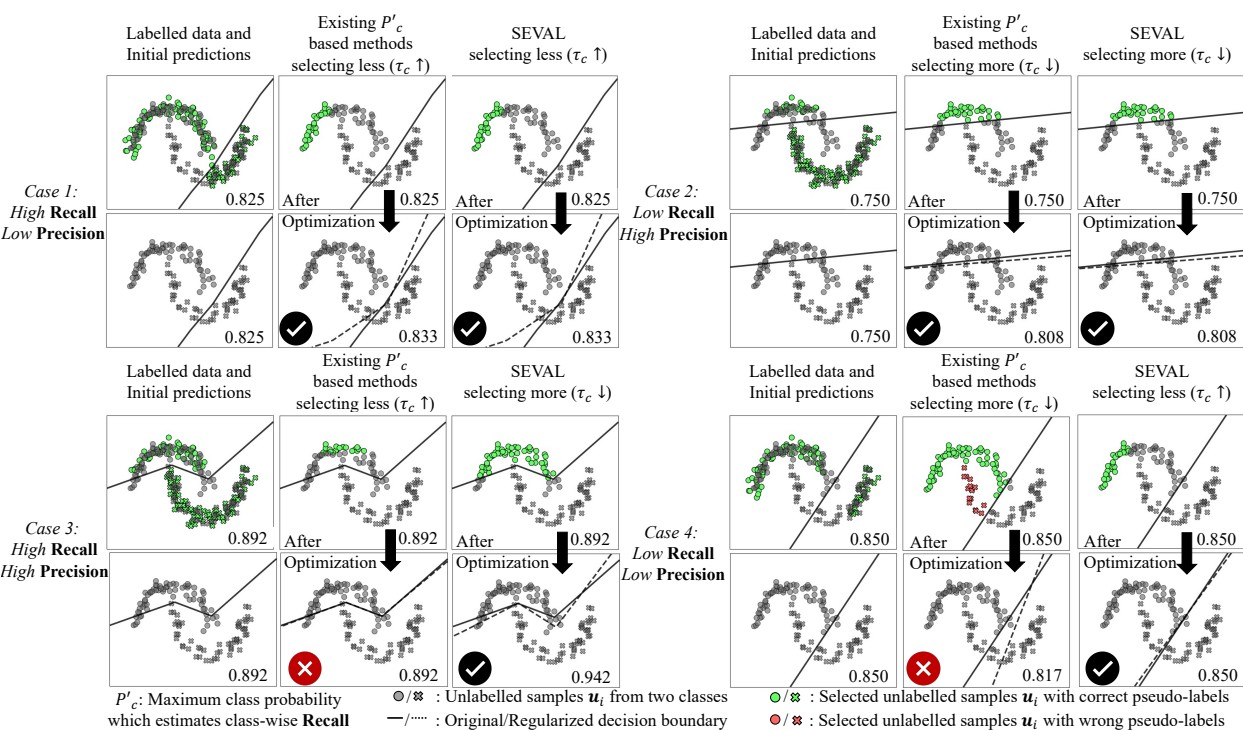

Figure 2: Two-moons toy experiments illustrating the relationship between threshold choice and model performance for class ◖. Accuracy appears in the bottom right. Current maximum class probability-based dynamic thresholding methods such as FlexMatch (Zhang et al., 2021), emphasizing **Recall**, may not be reliable for *Case 3* and *Case 4*. In comparison, SEVAL derived thresholds, reflecting **Precision**, fit all cases well (c.f. Section 4.2).

We argue that their strategies are based on the assumption that **Recall** and **Precision** are always in a trade-off relationship, which arises from adjusting decision boundaries—for example, lower **Recall** often leads to higher **Precision**. However, they would fall short if this does not hold. For example, we should choose as much as possible if the class is well-classified, e.g. high **Recall** and high **Precision**. However, following their strategy, they will choose few from them. We illustrate this in Fig. 2 with the two-moons example. While *Case 1* and *Case 2* are the most common scenarios, current maximum class probability-based approaches struggle to estimate thresholds effectively in other cases. We substantiate this assertion in the experimental section, where we find that *Case 3* frequently arises for the minority class in imbalanced SSL and is currently not adequately addressed, as shown in Section 5.6 and Appendix Section H.

Therefore, in this study, we propose to learn the thresholds that can reflect **Precision** and maximum $P_{\hat{\mathcal{U}}}(y = \hat{y})$ as stated in Eq. 5. We find that **Precision** is difficult to estimate directly from the network output because the network's probability does not inherently account for false negatives, which is essential for **Precision** 5.6. Therefore, similar to the approach used for PLR, we utilize a holdout dataset to estimate **Precision** and determine the thresholds based on this estimation. We detail the formulation in Section 4.2.

## 4 SEVAL

Fig. 3 shows an overview of SEVAL. The comparative advantage of SEVAL over existing methodologies is demonstrated in Table 1, Fig. 1, and Fig. 2. Critically, we propose to optimize both PLR and THA parameters based on our previous analysis, employing properly justified criteria to ensure optimal pseudo-label utilization in class-imbalanced SSL.

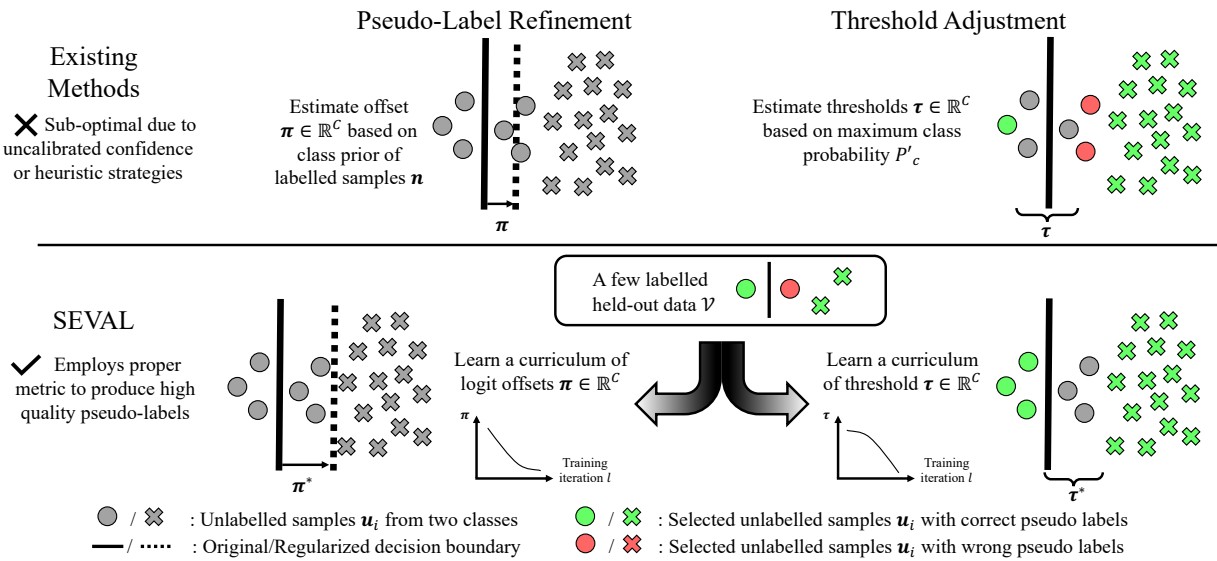

Figure 3: Overview of SEVAL optimization process which consists of two learning strategies aiming at mitigating bias in pseudo-labels within imbalanced SSL scenarios: PLR and THA. The parameter learning curriculum is determined by evaluating held-out data performance, which ensures greater accuracy while preventing overfitting. Note that PLR and THA are entirely complementary, as they address fundamentally different stages of the pseudo-labeling pipeline.

Independent of the training dataset $\mathcal{X}$ and $\mathcal{U}$, we assume we have access to a held-out dataset $\mathcal{V} = \{(\boldsymbol{x}_i, y_i)\}_{i=1}^K$, which contains $k_c$ samples for class $c$. We use this data to learn $\boldsymbol{\pi}$ for PLR and $\boldsymbol{\tau}$ for THA. We make no assumptions regarding $k_c$; that is, $\mathcal{V}$ can either be balanced or imbalanced [7].

### 4.1 Learning Pseudo-Label Refinement

#### 4.1.1 Learning Offsets

We want to further harness the potential of pseudo-label refinement by optimizing $\boldsymbol{\pi}$ from the data itself. We propose to directly estimate the optimal decision boundary as required in Proposition 1. As stated in Remark 1, under the assumption that $P^{\mathcal{T}}(Y)$ is uniform, we optimize the parameters by minimizing the cross-entropy loss averaged across classes. Specifically, the optimal offsets $\boldsymbol{\pi}$, are optimized using the labelled held-out data $\mathcal{V}$ with:

$$
\begin{aligned}
\boldsymbol{\pi}^* &= \arg\min_{\boldsymbol{\pi}} \sum_{j=1}^{C} \frac{1}{Ck_j} \sum_{i=1}^{K} \mathbb{1}(y_i = j)\mathcal{L}(y_i, \boldsymbol{p}_i^{\mathcal{V}}) \\
&= \arg\min_{\boldsymbol{\pi}} \sum_{j=1}^{C} \frac{1}{Ck_j} \sum_{i=1}^{K} \mathbb{1}(y_i = j)\mathcal{L}(y_i, \sigma(\boldsymbol{z}_i^{\mathcal{V}} - \log\boldsymbol{\pi})),
\end{aligned}
\tag{7}
$$

where $\mathcal{L}$ is the cross-entropy loss and $\boldsymbol{z}_i^{\mathcal{V}}$ is the network output for an input sample. Subsequently, we can compute the refined pseudo-label logit as $\hat{\boldsymbol{z}}_i^{\mathcal{U}} - \log\boldsymbol{\pi}^*$, which are expected to become more accurate on a class-wise basis. Of note, because Eq. 7 uses class-averaged cross-entropy loss, it avoids needing assumptions about the class-conditional likelihood for held-out data $\mathcal{V}$.

---

[7]In practice, $\mathcal{V}$ is normally separated from $\mathcal{X}$, c.f. Section 4.3. In implementation, we do not require a sizable $\mathcal{V}$, as we find that classes with similar class ratios will have similar offsets and thresholds, and thus their parameters can be optimized together at the group level, c.f. Section 4.2.4.

### 4.1.2 Extension to Test Logit Inference

As established in Proposition 1, the learned offsets $\boldsymbol{\pi}^*$ maintain optimality on test data with a uniform class distribution – a standard evaluation setting. Therefore, we expect $\boldsymbol{\pi}^*$ to not only improve pseudo-labeling on unlabelled data $\mathcal{U}$ but also outperform class frequency-based alternatives like LA (Zhou & Liu, 2005) during inference on the test set. This is because LA assumes the learned networks are calibrated to reflect the true probabilities of $P^{\mathcal{X}}(X|Y)$, which is rarely satisfied in practice.

Particularly, because the learned offsets are intrinsically linked to network parameters, we employ the final $\boldsymbol{\pi}^*$ learned on $\mathcal{V}$ to refine test results, anticipating superior performance compared to alternative methods like LA.

## 4.2 Learning Threshold Adjustment

### 4.2.1 Thresholding Criteria

For unlabeled samples, we select samples with largest predicted probabilities. A lower thresholds $\tau$ will lead to lower accuracy, but include more samples. As we demonstrated earlier in Remark 2, we should rely on **Precision** to determine class-specific thresholds since we can only access $\hat{y}$ but not $y$ for unlabeled samples. Specifically, we claim that (with the justification to follow later in the text):

**Remark 3** *When the number of selected unlabeled samples is fixed, the optimal threshold $\boldsymbol{\tau}$ to achieve the highest pseudo-label accuracy should ensure that every class in the selected samples has the same* ***Precision***.

We begin with Eq. 5, which maximizes pseudo-label correctness in our running example:

$$\max_{\boldsymbol{\tau}} \sum_{c=1}^{C} \mathcal{A}(\tau_c, c)\mathcal{S}(\tau_c, c), \quad \text{s.t.} \quad \sum_{c=1}^{C} \mathcal{S}(\tau_c, c) = M, \tag{8}$$

where $\boldsymbol{\tau}$ is the thresholds for sample selection, with its $c$'th element corresponding to the threshold for class $c$. $\mathcal{A}(\tau_c, c)$ represents **Precision** of class $c$ when selecting samples using $\tau_c$. Specifically, $\mathcal{A}(\tau_c, c)$ computes the accuracy of samples predicted as class $c$ whose maximum probability exceeds $\tau_c$:

$$\mathcal{A}(\tau_c, c) = \frac{1}{\mathcal{S}(\tau_c, c)} \sum_{i=1}^{K} \mathbb{1}_{ic} \mathbb{1}(y_i = c) \mathbb{1}(\max_{j}(p_{ij}^{\mathcal{V}}) > \tau_c), \tag{9}$$

where $\mathbb{1}_{ic}$ indicates if the most probable class is $c$ for samples $i$ as we mentioned earlier and $\mathcal{S}(\tau_c, c) = \sum_{i=1}^{K} \mathbb{1}_{ic} \mathbb{1}(\max_j(p_{ij}^{\mathcal{V}}) > \tau_c)$ is the number of samples predicted as class $c$ with confidence larger than $\tau_c$.

Using Lagrange multipliers, we formulate the Lagrangian for this problem with associated multiplier $\lambda$ as:

$$\Lambda(\tau_c, \lambda) = \sum_{c=1}^{C} \mathcal{A}(\tau_c, c)\mathcal{S}(\tau_c, c) - \lambda\Big(\mathcal{S}(\tau_c, c) - M\Big). \tag{10}$$

Taking the derivative of $\Lambda$ with respect to $\tau_c$ and setting it to zero yields the optimality conditions:

$$\frac{\partial}{\partial \tau_c}\Big[\mathcal{A}(\tau_c, c)\mathcal{S}(\tau_c, c)\Big] = \lambda\frac{\partial \mathcal{S}(\tau_c, c)}{\partial \tau_c}, \quad \forall c. \tag{11}$$

Since our focus is solely on the trade-off in sample selection across classes, we can adjust the thresholds $\tau_c$ to ensure that the selected samples are updated at a similar rate. In other words, it is safe to assume that $\frac{\partial \mathcal{S}(\tau_c, c)}{\partial \tau_c} = \lambda_{\mathcal{S}}$ across all classes, where $\lambda_{\mathcal{S}}$ is a constant.

Thus after some rearrangement, we can obtain:

$$\mathcal{A}(\tau_c, c)\lambda_{\mathcal{S}} + \mathcal{S}(\tau_c, c)\frac{\partial \mathcal{A}(\tau_c, c)}{\partial \tau_c} = \lambda\lambda_{\mathcal{S}}, \quad \forall c. \tag{12}$$

We further assume that the changes in accuracy are smooth and tend to be flatten, meaning that adjusting threshold $\tau_c$ does not significantly affect the overall accuracy of the specific class. This assumption is reasonable when a sufficient number of unlabeled samples are available per class. Under this condition, the term $\frac{\partial \mathcal{A}(\tau_c, c)}{\partial \tau_c}$ approaches 0, and consequently, $\epsilon = \mathcal{S}(\tau_c, c)\frac{\partial \mathcal{A}(\tau_c, c)}{\partial \tau_c}$ becomes very small. As a result, Eq. 12 can be approximated as:

$$\mathcal{A}(\tau_c, c) = \frac{\lambda\lambda_{\mathcal{S}} - \epsilon}{\lambda_{\mathcal{S}}}, \quad \forall c. \tag{13}$$

This demonstrates that at the optimal solution, $\mathcal{A}(\tau_c, c)$ for each class should be equal. This condition enforces a relationship between the thresholds $\tau_c$ across classes, ensuring a unique solution that satisfies Eq. 10. Consequently, we conclude that the optimal solution can be achieved by aligning the **Precision** across all classes to a common level. From a theoretical perspective, our criterion represents a multi-class analogue of the Neyman–Pearson Lemma (NPL). Additional analysis can be found in Appendix Section D.

### 4.2.2  Learning Thresholds

Building on this analysis, we optimize the thresholds to guarantee equal class-wise accuracy - specifically, the **Precision** metric from Eq. 5 - at target level $t$ for selected samples across all classes. As stated earlier, **Precision** cannot be determined solely by network logits $\boldsymbol{z}$, since it is also influenced by non-maximum probabilities (a point we further elucidate in Section 5.6). Thus, here we explore a method to learn optimal thresholds using an external held-out dataset $\mathcal{V}$, aiming to obtain pseudo-labels with maximized accuracy under given budgets. This is achieved by:

$$\tau_c^* = \begin{cases} \arg\min_{\tau_c} \left| \mathcal{A}(\tau_c, c) - t \right| & \text{if} \quad \alpha_c < t \\ 0 & \text{otherwise} \end{cases}, \tag{14}$$

where $\alpha_c = \frac{1}{K_c}\sum_{i=1}^{K} \mathbb{1}_{ic}\mathbb{1}(y_i = c)$ is the average accuracy of all the samples predicted as class $c$. Notably, when $\alpha_c$ exceeds threshold $t$, we think the pseudo-labels are sufficiently accurate to include all instances.

The thresholds optimized with Eq. 14 are inversely related to **Precision** and possess practical utility in handling classes with varying accuracy. We believe this cost function is better suited for fair threshold optimization across diverse class difficulties. This optimization approach introduces no additional hyperparameters, simply replacing the previous $\tau$ with $t$.

In practical scenarios, we could face difficulties in directly determining the threshold through Eq. 14 due to the imbalances in held-out data and constraints arising from a limited sample size. We address these issues via normalized cost functions and group-based optimization, defined as follows.

### 4.2.3  Learning with Imbalanced Held-Out Data

As the labelled training dataset $\mathcal{X}$ is imbalanced, in practice, it is hard to obtain a balanced split $\mathcal{V}$ to learn a curriculum of threshold $\boldsymbol{\tau}$. However, when we optimize $\boldsymbol{\tau}$ using an imbalanced validation $\mathcal{V}$ following Eq. 14, the optimized results would be biased. More precisely, the majority class consistently exhibits high **Precision**, leading to a lower threshold, while the opposite holds true for the minority class. Therefore, we utilize the class frequency of the labelled held-out data $\boldsymbol{k}$ to normalize the cost function. Specifically, we calculate the class weight as $\boldsymbol{\omega}^{\mathcal{V}} = 1/\boldsymbol{k}$. This parameter would assign large weight to the minority class and small weight to the majority classes. Then we replace all the $\mathbb{1}_{ic}$ with $\omega_{y_i}^{\mathcal{V}}\mathbb{1}_{ic}$ in Eq. 14, obtaining:

$$\tau_c^* = \begin{cases} \underset{\tau_c}{\arg\min} \left| \frac{1}{\mathcal{S}(\tau_c, c)} \sum_{i=1}^{K} \omega_{y_i}^{\mathcal{V}} \mathbb{1}_{ic} \mathbb{1}(y_i = c) \mathbb{1}\left( \max_j (p_{ij}^{\mathcal{V}}) > \tau_c \right) - t \right| & \text{if } \alpha_c < t \\ 0 & \text{otherwise} \end{cases}, \qquad (15)$$

where $\mathcal{S}(\tau_c, c) = \sum_{i=1}^{K} \omega_{y_i}^{\mathcal{V}} \mathbb{1}_{ic} \mathbb{1}(\max_j(p_{ij}^{\mathcal{V}}) > \tau_c)$ is the normalized number of samples predicted as class $c$ with confidence larger than $\tau_c$, where $\alpha_c = \frac{1}{K_c} \sum_{i=1}^{K} \omega_{y_i}^{\mathcal{V}} \mathbb{1}_{ic} \mathbb{1}(y_i = c)$ is the average balanced accuracy of all the samples predicted as class $c$ and $K_c = \sum_{i=1}^{K} \omega_{y_i}^{\mathcal{V}} \mathbb{1}_{ic}$ is the normalized number of samples predicted as $c$. This modification can normalize the number of samples within the cost function. Consequently, we can directly learn the thresholds $\boldsymbol{\tau}$ using imbalanced held-out data. This is a vital element for SEVAL in real-world SSL, where imbalanced data is the norm and fully labeled, balanced datasets are scarce.

### 4.2.4 Learning Thresholds within Groups

When we learn $\boldsymbol{\tau}$ based on the held-out data $\mathcal{V}$, the optimization process could be unstable as sometimes we have very few samples per class (e.g. less than 10 samples). In this case, even if we can re-weight the validation samples based on their class prior $\boldsymbol{k}$, it is hard to have enough samples to obtain stable $\boldsymbol{\tau}$ curriculum for the minority classes, especially when $\min_c(k_c) < 10$. Assuming equal class priors should result in similar thresholds, we propose to optimize thresholds within groups, pinpointing the ideal ones that fulfill the accuracy requirement for every classes within the group.

We assume the samples of different classes $k_c$ are arranged in descending order. In other words, $k_1$ is the maximum, and $k_C$ is the minimum. Instead of optimizing $\tau_c$ for an individual class $c$, we optimize for groups such that the learned $\tilde{\tau}_b$ can satisfy the accuracy requirements for $B$ classes. Specifically, the optimal $\tilde{\boldsymbol{\tau}} \in \mathbb{R}^{\lceil C/B \rceil}$ is determined as:

$$\tilde{\tau}_b^* = \begin{cases} \underset{\tilde{\tau}_b}{\arg\min} \left| \frac{1}{\tilde{\mathcal{S}}(\tilde{\tau}_b, b)} \sum_{c=bB+1}^{bB+B} \sum_{i=1}^{K} \mathbb{1}_{ic} \mathbb{1}(y_i = c) \mathbb{1}(\max_j(p_{ij}^{\mathcal{V}}) > \tilde{\tau}_b) - t \right| & \text{if } \tilde{\alpha}_b < t \\ 0 & \text{otherwise} \end{cases}, \qquad (16)$$

where $\tilde{\mathcal{S}}(\tilde{\tau}_b, b) = \sum_{c=bB+1}^{bB+B} \sum_{i=1}^{K} \mathbb{1}_{ic} \mathbb{1}(\max_j(p_{ij}^{\mathcal{V}}) > \tilde{\tau}_b)$ is the number of samples that are chosen in this group based on the threshold $\tilde{\tau}_b$ and $\tilde{\alpha}_b = \frac{1}{\sum_{c=bB+1}^{bB+B} K_c} \sum_{c=bB+1}^{bB+B} \sum_{i=1}^{K} \mathbb{1}_{ic} \mathbb{1}(y_i = c)$ is the average accuracy of all the samples predicted as class in this group. If we set $B = 1$, Eq. 16 becomes equivalent to Eq. 14. The necessity of the group-based optimization depends on data availability: it is essential when labeled data per class is scarce ($k_c < 10$) to guarantee a minimum group size, but less critical with sufficient labeled samples. Additionally, it mitigates threshold instability arising from a poor base model that fails to produce positive class predictions (see Appendix Section G).

## 4.3 Curriculum Learning

After obtaining the optimal refinement parameters, for pseudo-label $\hat{y}_i = \arg\max_j(q_{ij})$ and predicted class $y_i' = \arg\max_j(p_{ij}^{\mathcal{U}})$, we can calculate the unlabelled loss $\hat{R}_{\mathcal{L}, \hat{\mathcal{U}}}(f) = \frac{1}{M} \sum_{i=1}^{M} \mathbb{1}(\max_j(q_{ij}) \geq \tau_{y_i'}) \mathcal{L}(\hat{y}_i, \boldsymbol{p}_i^{\mathcal{U}})$ to update our classification model parameters. The estimation process of $\boldsymbol{\pi}$ and $\boldsymbol{\tau}$ is summarized in Appendix Algorithm 2.

To bypass more data collection efforts, we learn the curriculum of $\boldsymbol{\pi}$ and $\boldsymbol{\tau}$ based on a partition of labelled training dataset $\mathcal{X}$ thus *we do not require additional samples*. Specifically, $\mathcal{X}$ is partitioned into two equally sized subsets $\mathcal{X}'$ and $\mathcal{V}'$ prior to SSL training, from which the $L$-parameter curriculum is learned. As $\boldsymbol{\pi}$ and $\boldsymbol{\tau}$ address complementary aspects of the problem, their optimization can in principle be parallelized with network training, incurring at most 50% theoretical overhead since the curriculum is learned on only half the training data. We expect this overhead to be further reducible, as the training dynamics remain consistent across different learning schedules.

In order to ensure curriculum stability, we update the parameters with exponential moving average. Specifically, when we learn a curriculum of length $L$, after several iterations, we optimize $\boldsymbol{\pi}$ and $\boldsymbol{\tau}$ sequentially based on current model status. We then calculate the curriculum for step $l$ as $\boldsymbol{\pi}^{(l)} = \eta_\pi \boldsymbol{\pi}^{(l-1)} + (1 - \eta_\pi)\boldsymbol{\pi}^{(l)*}$ and $\boldsymbol{\tau}^{(l)} = \eta_\tau \boldsymbol{\tau}^{(l-1)} + (1 - \eta_\tau)\boldsymbol{\tau}^{(l)*}$. We use this to refine pseudo-label before the next SEVAL parameter update. We initialize with FixMatch's thresholds (Sohn et al., 2020) and gradually transition to optimized thresholds to ensure class-wise pseudo-label accuracy exceeds $t$. We summarize the training process of SEVAL in Appendix Algorithm 1.

## 5 Experiments

We conducted main experiments on several imbalanced SSL benchmarks including CIFAR-10-LT, CIFAR-100-LT (Krizhevsky et al., 2009), STL-10-LT (Coates et al., 2011) and ImageNet-127 (Deng et al., 2009) under the same codebase, following (Oh et al., 2022). The training datasets are intentionally resampled from their original balanced versions using an exponential distribution to create class imbalance. We further apply SEVAL to the realistic imbalanced SSL dataset, Semi-Aves (Su & Maji, 2021), which is a dataset for bird species recognition, exhibiting natural class imbalance where some species have over 100 images while others have fewer than 10. To compute the curriculum for $\boldsymbol{\pi}$ and $\boldsymbol{\tau}$, we split $\mathcal{X}$ equally into $\mathcal{X}'$ and $\mathcal{V}'$, resulting in $\boldsymbol{k} = \boldsymbol{n}/2$ validation samples per class.

We assume a uniform test label distribution $P^{\mathcal{T}}(Y)$, which aligns with standard test dataset construction and evaluation practices in the literature. For all datasets except ImageNet-127, the test sets are constructed with an equal number of samples per class, and we report accuracy as the primary evaluation metric. For ImageNet-127, where the test dataset is inherently imbalanced, we instead report the averaged class-wise **Recall** to ensure fair performance evaluation.

We choose wide ResNet-28-2 (Zagoruyko & Komodakis, 2016) as the feature extractor and train the network at a resolution of $32 \times 32$. We train the neural networks for 250,000 iterations with fixed learning rate of 0.03. We control the imbalance ratios for both labelled and unlabelled data ($\gamma^{\mathcal{X}}$ and $\gamma^{\mathcal{U}}$) and exponentially decrease the number of samples per class. We set the curriculum length $L$ to 500 for datasets with fewer classes and approximately 100 for those with more classes. Full implementation details and hyperparameter settings are provided in Appendix Section G.

For most experiments, we employ FixMatch to calculate the pseudo-label and make the prediction using the exponential moving average version of the model following (Sohn et al., 2020). We report the average test accuracy along with its variance, derived from three distinct random seeds. These random seeds control both dataset splitting and the initialization of optimization algorithms. We ensure rigorous fairness in comparisons by conducting *all* reported experiments using identical codebase implementations under controlled conditions.

### 5.1 Main Results

We compared SEVAL with different SSL algorithms and summarize the test accuracy results in Table 2. To ensure fair comparison of algorithm performance, in this table, we mark SSL algorithms based on the way they tackle the imbalance challenge. In particular, techniques such as DARP, which exclusively manipulate the probability of pseudo-labels $\boldsymbol{\pi}$, are denoted as PLR. In contrast, approaches like FlexMatch, which solely alter the threshold $\boldsymbol{\tau}$, are termed as THA. We denote other methods that apply regularization techniques to the model's cost function using labelled data as long-tailed learning (LTL). In addition to SEVAL results, we also report the results of SEVAL-PL, which forgoes any post-hoc adjustments on test samples. This ensures that its results are directly comparable with its counterparts. Here we pair SEVAL-PL with the simplest LTL strategy, namely optimized post-hoc LA. Although SEVAL can readily be combined with other LTL approaches, such as data reweighting, to yield further gains, this work focuses on the core challenges of imbalanced SSL: PLR and THA. We therefore emphasize the effectiveness of SEVAL-PL and leave further engineering extensions to future work.

As shown in Table 2, SEVAL-PL outperforms other PLR and THA based methods such as DARP, FlexMatch and FreeMatch with a considerable margin. This indicates that SEVAL can provide better pseudo-label for

Table 2: Accuracy on CIFAR10-LT, CIFAR100-LT and STL10-LT. We divide SSL algorithms into different groups including long-tailed learning (LTL), pseudo-label refinement (PLR) and threshold adjustment (THA). PLR and THA based methods only modify pseudo-label probability $q_i$ and threshold $\tau$, respectively. Best results within the same category are in **bold** for each configuration.

| Algorithm | Method type | | | CIFAR10-LT $\gamma^{\mathcal{X}} = \gamma^{\mathcal{U}} = 100$ | | CIFAR100-LT $\gamma^{\mathcal{X}} = \gamma^{\mathcal{U}} = 10$ | | STL10-LT $\gamma^{\mathcal{X}} = 20, \gamma^{\mathcal{U}}$: unknown | |
|---|---|---|---|---|---|---|---|---|---|
| | LTL | PLR | THA | $n_1 = 500$ $m_1 = 4000$ | $n_1 = 1500$ $m_1 = 3000$ | $n_1 = 50$ $m_1 = 400$ | $n_1 = 150$ $m_1 = 300$ | $n_1 = 150$ $M = 100,000$ | $n_1 = 450$ |
| Supervised | | | | $47.3 \pm 0.95$ | $61.9 \pm 0.41$ | $29.6 \pm 0.57$ | $46.9 \pm 0.22$ | $39.4 \pm 1.40$ | $51.7 \pm 2.21$ |
| w/ LA (Menon et al., 2020) | ✓ | | | $53.3 \pm 0.44$ | $70.6 \pm 0.21$ | $30.2 \pm 0.44$ | $48.7 \pm 0.89$ | $42.0 \pm 1.24$ | $55.8 \pm 2.22$ |
| FixMatch (Sohn et al., 2020) | | | | $67.8 \pm 1.13$ | $77.5 \pm 1.32$ | $45.2 \pm 0.55$ | $56.5 \pm 0.06$ | $47.6 \pm 4.87$ | $64.0 \pm 2.27$ |
| w/ DARP (Kim et al., 2020) | | ✓ | | $74.5 \pm 0.78$ | $77.8 \pm 0.63$ | $49.4 \pm 0.20$ | $58.1 \pm 0.44$ | $59.9 \pm 2.17$ | $72.3 \pm 0.60$ |
| w/ FlexMatch (Zhang et al., 2021) | | | ✓ | $74.0 \pm 0.64$ | $78.2 \pm 0.45$ | $49.9 \pm 0.61$ | $58.7 \pm 0.24$ | $48.3 \pm 2.75$ | $66.9 \pm 2.34$ |
| w/ Adsh (Guo & Li, 2022) | | | ✓ | $73.0 \pm 3.46$ | $77.2 \pm 1.01$ | $49.6 \pm 0.64$ | $58.9 \pm 0.71$ | $60.0 \pm 1.75$ | $71.4 \pm 1.37$ |
| w/ FreeMatch (Wang et al., 2022d) | | ✓ | ✓ | $73.8 \pm 0.87$ | $77.7 \pm 0.23$ | $49.8 \pm 1.02$ | $59.1 \pm 0.59$ | $63.5 \pm 2.61$ | $73.9 \pm 0.48$ |
| w/ SEVAL-PL | | ✓ | ✓ | $\mathbf{77.7} \pm 1.38$ | $\mathbf{79.7} \pm 0.53$ | $\mathbf{50.8} \pm 0.84$ | $\mathbf{59.4} \pm 0.08$ | $\mathbf{67.4} \pm 0.79$ | $\mathbf{75.2} \pm 0.48$ |
| w/ ABC (Lee et al., 2021) | ✓ | | | $78.9 \pm 0.82$ | $83.8 \pm 0.36$ | $47.5 \pm 0.18$ | $59.1 \pm 0.21$ | $58.1 \pm 2.50$ | $74.5 \pm 0.99$ |
| w/ SAW (Lai et al., 2022b) | ✓ | | | $74.6 \pm 2.50$ | $80.1 \pm 1.12$ | $45.9 \pm 1.85$ | $58.2 \pm 0.18$ | $62.4 \pm 0.86$ | $74.0 \pm 0.28$ |
| w/ CReST+ (Wei et al., 2021) | ✓ | ✓ | | $76.3 \pm 0.86$ | $78.1 \pm 0.42$ | $44.5 \pm 0.94$ | $57.1 \pm 0.65$ | $56.0 \pm 3.19$ | $68.5 \pm 1.88$ |
| w/ DASO (Oh et al., 2022) | ✓ | ✓ | | $76.0 \pm 0.37$ | $79.1 \pm 0.75$ | $49.8 \pm 0.24$ | $59.2 \pm 0.35$ | $65.7 \pm 1.78$ | $75.3 \pm 0.44$ |
| w/ SimPro (Du et al., 2024a) | ✓ | ✓ | | $68.4 \pm 1.93$ | $78.8 \pm 0.10$ | $44.4 \pm 0.20$ | $56.2 \pm 0.20$ | $55.3 \pm 1.68$ | $67.1 \pm 1.64$ |
| w/ MetaExpert (Hou et al., 2025) | ✓ | ✓ | | $80.9 \pm 0.89$ | $84.4 \pm 0.60$ | $48.9 \pm 0.70$ | $58.2 \pm 0.64$ | $62.4 \pm 0.69$ | $75.6 \pm 0.27$ |
| w/ ACR (Wei & Gan, 2023) | ✓ | ✓ | ✓ | $80.2 \pm 0.78$ | $83.8 \pm 0.13$ | $50.6 \pm 0.13$ | $60.7 \pm 0.23$ | $65.6 \pm 0.11$ | $\mathbf{76.3} \pm 0.57$ |
| w/ SEVAL | ✓ | ✓ | ✓ | $\mathbf{82.8} \pm 0.56$ | $\mathbf{85.3} \pm 0.25$ | $\mathbf{51.4} \pm 0.95$ | $\mathbf{60.8} \pm 0.28$ | $\mathbf{67.4} \pm 0.69$ | $75.7 \pm 0.36$ |

the models by learning a better curriculum for $\boldsymbol{\pi}$ and $\boldsymbol{\tau}$. When compared with other hybrid methods including ABC, CReST+, DASO, ACR, SEVAL demonstrates significant advantages in most scenarios. Relying solely on the strength of pseudo-labeling, SEVAL delivers highly competitive performance in the realm of imbalanced SSL. Given its straightforward framework, SEVAL can be integrated with other SSL concepts to enhance accuracy, a point we delve into later in the ablation study.

Semi-Aves contains 200 classes with different long-tailed distribution and captures a situation where a portion of the unlabelled data originates from previously unseen classes. In addition to labelled data, Semi-Aves also contains imbalanced unlabelled data $\mathcal{U}_{\text{in}}$ and unlabelled open-set data $\mathcal{U}_{\text{out}}$ from another 800 classes. Following previous works (Su et al., 2021; Oh et al., 2022), we conducted experiments using $\mathcal{U}_{\text{in}}$ or a combination of $\mathcal{U}_{\text{in}}$ and $\mathcal{U}_{\text{out}}$. We summarize the results in Table 3. This dataset poses a challenge due to the limited number of samples in the tail class, with only around 15 samples per class. It has been observed that SEVAL performs effectively in such a demanding scenario. We additionally present a summary of further experimental results under various realistic settings, along with sensitivity analyses, in Appendix F.

Table 3: Accuracy on Semi-Aves. Best results within the same category are in **bold** for each configuration. The training set has severe class imbalance, with sample counts per class ranging from $n_1 = 53$ to $n_{200} = 15$.

| Algorithm | Method type | | | Semi-Aves | |
|---|---|---|---|---|---|
| | LTL | PLR | THA | $\mathcal{U} = \mathcal{U}_{\text{in}}$ | $\mathcal{U} = \mathcal{U}_{\text{in}} + \mathcal{U}_{\text{out}}$ |
| FixMatch (Sohn et al., 2020) | | | | $59.9 \pm 0.08$ | $52.6 \pm 0.14$ |
| w/ DARP (Kim et al., 2020) | | ✓ | | $60.3 \pm 0.24$ | $54.7 \pm 0.06$ |
| w/ SEVAL-PL | | ✓ | ✓ | $\mathbf{60.6} \pm 0.18$ | $\mathbf{56.4} \pm 0.10$ |
| w/ CReST+ (Wei et al., 2021) | ✓ | ✓ | | $60.0 \pm 0.03$ | $54.3 \pm 0.59$ |
| w/ DASO (Oh et al., 2022) | ✓ | ✓ | | $59.3 \pm 0.28$ | $56.6 \pm 0.32$ |
| w/ SEVAL | ✓ | ✓ | ✓ | $\mathbf{60.7} \pm 0.17$ | $\mathbf{56.7} \pm 0.15$ |

Table 4: Accuracy on CIFAR10-LT with different imbalanced ratios. Best results within the same category are in **bold** for each configuration.

| | Method type | | | CIFAR10-LT | | | | | |
| | | | | $\gamma^{\mathcal{X}} = 100, \gamma^{\mathcal{U}} = 1$ | | $\gamma^{\mathcal{X}} = 100, \gamma^{\mathcal{U}} = 1/100$ | | $\gamma^{\mathcal{X}} = \gamma^{\mathcal{U}} = 150$ | |
| Algorithm | LTL | PLR | THA | $n_1 = 500$ $m_1 = 4000$ | $n_1 = 1500$ $m_1 = 3000$ | $n_1 = 500$ $m_1 = 4000$ | $n_1 = 1500$ $m_1 = 3000$ | $n_1 = 500$ $m_1 = 4000$ | $n_1 = 1500$ $m_1 = 3000$ |
|---|---|---|---|---|---|---|---|---|---|
| FixMatch (Sohn et al., 2020) | | | | $73.0 \pm 3.81$ | $81.5 \pm 1.15$ | $62.5 \pm 0.94$ | $71.8 \pm 1.70$ | $62.9 \pm 0.36$ | $72.4 \pm 1.03$ |
| w/ DARP (Kim et al., 2020) | | ✓ | | $82.5 \pm 0.75$ | $84.6 \pm 0.34$ | $70.1 \pm 0.22$ | $80.0 \pm 0.93$ | $67.2 \pm 0.32$ | $73.6 \pm 0.73$ |
| w/ SEVAL-PL | | ✓ | ✓ | $\mathbf{89.4} \pm 0.53$ | $\mathbf{89.2} \pm 0.02$ | $\mathbf{77.7} \pm 0.91$ | $\mathbf{80.9} \pm 0.66$ | $\mathbf{71.9} \pm 1.10$ | $\mathbf{74.7} \pm 0.63$ |
| w/ CReST+ (Wei et al., 2021) | ✓ | ✓ | | $82.2 \pm 1.53$ | $86.4 \pm 0.42$ | $62.9 \pm 1.39$ | $72.9 \pm 2.00$ | $67.5 \pm 0.45$ | $73.7 \pm 0.34$ |
| w/ DASO (Oh et al., 2022) | ✓ | ✓ | | $86.6 \pm 0.84$ | $88.8 \pm 0.59$ | $71.0 \pm 0.95$ | $80.3 \pm 0.65$ | $70.1 \pm 1.81$ | $75.1 \pm 0.77$ |
| w/ SEVAL | ✓ | ✓ | ✓ | $\mathbf{90.3} \pm 0.61$ | $\mathbf{90.6} \pm 0.47$ | $\mathbf{79.2} \pm 0.83$ | $\mathbf{82.9} \pm 1.78$ | $\mathbf{79.8} \pm 0.42$ | $\mathbf{83.3} \pm 0.40$ |

## 5.2 Varied Imbalanced Ratios

Similar to results in Table 2, we evaluate SEVAL on CIFAR10-LT with different imbalanced ratios and summarize results in Table 4. We find that SEVAL consistently outperforms its counterparts across different $\gamma^{\mathcal{X}}$ values. Since SEVAL does not make any assumptions about the distribution of unlabeled data, it can be robustly implemented in scenarios where $\gamma^{\mathcal{X}} \neq \gamma^{\mathcal{U}}$. In these settings, SEVAL's performance advantage over its counterparts is even more pronounced.

## 5.3 Low Labelled Data Scheme

Table 5: Accuracy on CIFAR10-LT under the setting of extremely few labelled samples. Best results within the same category are in **bold** for each configuration.

| | Method type | | | CIFAR10-LT | | |
| | | | | $\gamma^{\mathcal{X}} = \gamma^{\mathcal{U}} = 100$ | $\gamma^{\mathcal{X}} = 1, \gamma^{\mathcal{U}} = 100$ | |
| Algorithm | LTL | PLR | THA | $n_1 = 200$ $m_1 = 4000$ | $n_1 = 10$ $m_1 = 4000$ | $n_1 = 4$ $m_1 = 4000$ |
|---|---|---|---|---|---|---|
| FixMatch (Sohn et al., 2020) | | | | $64.3 \pm 0.83$ | $65.3 \pm 0.80$ | $44.7 \pm 3.33$ |
| w/ FreeMatch (Wang et al., 2022d) | | ✓ | ✓ | $67.4 \pm 1.09$ | $58.4 \pm 0.76$ | $50.7 \pm 1.95$ |
| w/ SEVAL-PL | | ✓ | ✓ | $\mathbf{69.3} \pm 0.66$ | $\mathbf{68.3} \pm 0.56$ | $\mathbf{51.5} \pm 1.51$ |
| w/ DASO (Oh et al., 2022) | ✓ | ✓ | | $67.2 \pm 1.25$ | $61.2 \pm 0.96$ | $48.6 \pm 2.81$ |
| w/ SEVAL | ✓ | ✓ | ✓ | $\mathbf{71.2} \pm 0.80$ | $\mathbf{68.9} \pm 0.25$ | $\mathbf{52.7} \pm 1.83$ |

SEVAL acquires a curriculum of parameters by partitioning the training dataset. This raises a crucial question: can SEVAL remain effective with a very limited number of labeled samples? To explore this, we conduct a stress test by training SEVAL with a minimal amount of labeled data.

In the first experimental configuration, we keep the imbalance ratio constant while reducing the number of labeled samples ($n_1 = 200$). In this extreme case, only two samples are labeled for the tail class. In the second configuration, we use a balanced labeled training dataset, but with a total of 100 and 40 samples for training. The results are summarized in Table 5. We find that SEVAL performs well in both scenarios, demonstrating robustness even when labeled data is scarce. Although reserving a held-out subset reduces the effective training set, the curriculum parameters $\boldsymbol{\pi}$ and $\boldsymbol{\tau}$ learned on this partition generalize well to the full training data.

## 5.4 Performance Analysis

To closely examine the distinct contributions of $\boldsymbol{\pi}$ and $\boldsymbol{\tau}$, we carry out an ablation study where SEVAL optimizes just one of them, respectively termed SEVAL-PLR and SEVAL-THA. Note that FixMatch's default setting maintains a constant $\pi$ of 1.0 and $\tau$ of 0.95 for all classes. As summarized in Table 6, SEVAL-PLR

Table 6: Comparison of SEVAL when only optimizing $\pi$ (SEVAL-PLR) or only optimizing $\tau$ (SEVAL-THA). SEVAL outperforms counterparts with identical parameter settings under different imbalanced SSL scenarios. SEVAL-PL, with its sequential optimization of both $\pi$ and $\tau$, yields further improvements in accuracy.

| | Method type | | | CIFAR10-LT | CIFAR100-LT |
|---|---|---|---|---|---|
| Algorithm | LTL | PLR | THA | $n_1 = 500$ $m_1 = 4000$ $\gamma^{\mathcal{X}} = \gamma^{\mathcal{U}} = 100$ | $n_1 = 150$ $m_1 = 300$ $\gamma^{\mathcal{X}} = \gamma^{\mathcal{U}} = 10$ |
| FixMatch (Sohn et al., 2020) | | | | $67.8 \pm 1.13$ | $56.5 \pm 0.06$ |
| w/ DARP (Kim et al., 2020) | | ✓ | | $74.5 \pm 0.78$ | $58.1 \pm 0.44$ |
| w/ SEVAL-PLR | | ✓ | | $76.7 \pm 0.82$ | $59.3 \pm 0.30$ |
| w/ FlexMatch (Zhang et al., 2021) | | | ✓ | $74.0 \pm 0.64$ | $58.7 \pm 0.24$ |
| w/ SEVAL-THA | | | ✓ | $77.0 \pm 0.93$ | $59.1 \pm 0.18$ |
| w/ SEVAL-PL | | ✓ | ✓ | $\mathbf{77.7} \pm 1.38$ | $\mathbf{59.4} \pm 0.08$ |

and SEVAL-THA can still outperform their counterparts, DARP and FlexMatch, respectively. When tuning both parameters, SEVAL-PL can achieve the best results.

### 5.4.1 Pseudo-Label Refinement

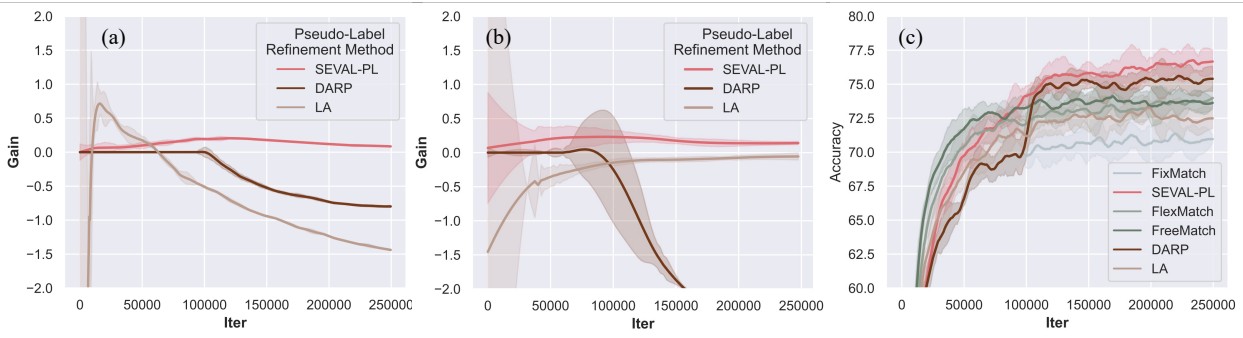

Figure 4: Evolution of (a,b) **Gain** and (c) test accuracy over training iterations. On CIFAR10-LT with $n_1 = 500$ (a) and CIFAR100-LT with $n_1 = 50$ (b), SEVAL achieves higher pseudo-label accuracy than baselines. In (c), on CIFAR10-LT with $n_1 = 500$, SEVAL-PL outperforms alternative SSL methods.

In order to comprehensively and quantitatively investigate the accuracy of pseudo-label refined by different approaches, here we define $G$ as the sum of accuracy gain and balanced accuracy gain of pseudo-label over training iterations. Both sample-wise accuracy and class-wise accuracy are crucial measures for evaluating the quality of pseudo-labels. A low sample-specific accuracy can lead to noisier pseudo-labels, adversely affecting model performance. Meanwhile, a low class-specific accuracy often indicates a bias towards the dominant classes. Therefore, we propose a metric $G$ which is the combination of these two metrics. Specifically, given the pseudo-label $\hat{y}_i$ and predicted class $y_i'$ of unlabelled dataset $\mathcal{U}$, we calculate $G$ as:

$$G = \underbrace{\frac{\sum_{i=1}^{M}[\mathbb{1}(\hat{y}_i' = y_i) - \mathbb{1}(\hat{y}_i = y_i)]}{M}}_{\text{Sample-Wise Accuracy Gain}} + \underbrace{\sum_{c=1}^{C}\sum_{i=1}^{M}\frac{\mathbb{1}(\hat{y}_i' = c)\mathbb{1}(\hat{y}_i' = y_i) - \mathbb{1}(\hat{y}_i = c)\mathbb{1}(\hat{y}_i = y_i)}{m_c C}}_{\text{Class-Wise Accuracy Gain}}. \tag{17}$$

To evaluate the cumulative impact of pseudo-labels, we calculate **Gain**(**iter**) as the accuracy gain at training iteration **iter** and monitor **Gain**(**iter**) $= \sum_{j=1}^{\textbf{iter}} G(j)/\textbf{iter}$ throughout the training iterations. The results of SEVAL along with DARP and adjusting pseudo-label logit $\hat{z}_c^{\mathcal{U}}$ with LA are summarized in Fig. 4(a). We note that SEVAL consistently delivers a positive **Gain** throughout the training iterations. In contrast, DARP and LA tend to reduce the accuracy of pseudo-labels during the later stages of the training process. After a warm-up period, DARP adjusts the distribution of pseudo-labels to match the inherent distribution of unlabelled data. However, it doesn't guarantee the accuracy of the pseudo-labels, thus not optimal. While LA can enhance class-wise accuracy, it isn't always the best fit for every stage of the model's learning. Consequently, noisy pseudo-labels from the majority class can impede the model's training. SEVAL learns a smooth curriculum of parameters for pseudo-label refinement from the data itself, therefore bringing more stable improvements. The test accuracy curves in Fig. 4(c) provide further validation of SEVAL's effectiveness, with SEVAL-PL demonstrating superior performance over both LA and DARP.

### 5.4.2 Threshold Adjustment

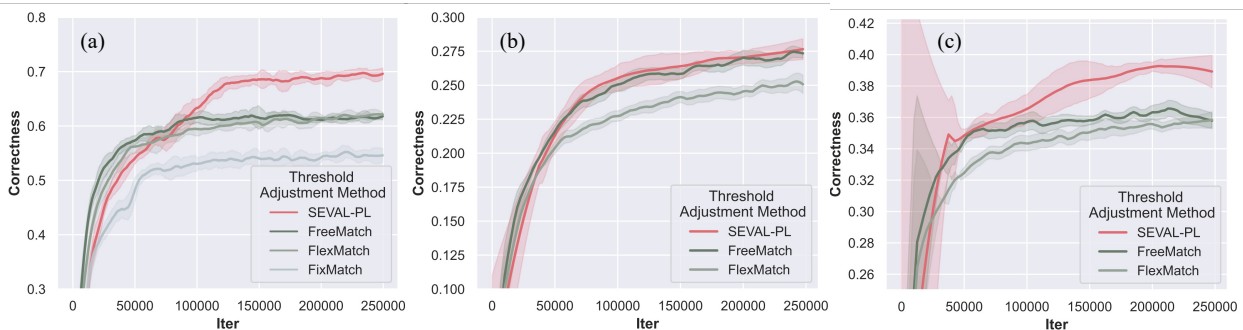

Figure 5: **Correctness** progression across training iterations. SEVAL exhibits an improved trade-off between quality and quantity. Results are presented on (a) CIFAR10-LT with $n_1 = 500$, (b) CIFAR100-LT with $n_1 = 50$, and (c) CIFAR100-LT with $n_1 = 150$.

Quantity and quality are two essential factors for pseudo-labels, as highlighted in (Chen et al., 2023). Quantity refers to the number of correctly labeled samples produced by pseudo-label algorithms, while quality indicates the proportion of correctly labeled samples after applying confidence-based thresholding.

In order to access the effectiveness of pseudo-label, we propose a metric called **Correctness**, which is a combination of quantity and quality. Having just high quantity or just high quality isn't enough for effective pseudo-labels. For instance, setting exceedingly high thresholds might lead to the selection of a limited number of accurately labelled samples (high quality). However, this is not always the ideal approach, and the opposite holds true for quantity. Therefore, the proposed **Correctness** metric combines both quality and quantity for fair evaluation. In particular, factoring in the potential imbalance of unlabelled data, we utilize a class frequency based weight term $\boldsymbol{\omega}^{\mathcal{U}} = 1/\boldsymbol{m}$ to normalize this metric, yielding:

$$\textbf{Correctness} = \frac{\mathcal{C}}{\underbrace{\sum_{i=1}^{M} \omega_{y_i}^{\mathcal{U}}}_{\text{Quantity}}} \underbrace{\frac{\mathcal{C}}{\sum_{i=1}^{M} \omega_{y_i}^{\mathcal{U}} \mathbb{1}(\max_j(q_{ij}) \geq \tau_{y_i'})}}_{\text{Quality}}, \tag{18}$$

where, $\mathcal{C} = \sum_{i=1}^{M} \omega_{y_i}^{\mathcal{U}} \mathbb{1}(\hat{y}_i = y_i)\mathbb{1}(\max_j(q_{ij}) \geq \tau_{y_i'})$ is the relative number of correctly labelled samples. We show **Correctness** of SEVAL with FixMatch, FlexMatch and FreeMatch in Fig. 5. We observe that FlexMatch and FreeMatch can both improve **Correctness**, while SEVAL can boost even more. We observe that the test accuracy follows a trend similar to **Correctness**, as shown in Fig. 5(a) and Fig. 4(c). This demonstrates that the thresholds set by SEVAL not only ensure a high quantity but also attain high accuracy for pseudo-labels, making them efficient in the model's learning process. The correlation between **Correctness** and test accuracy suggests that **Correctness**

could serve as a practical proxy for dynamically tuning the hyperparameter $t$. It could be achieved by adjusting it each epoch to maximize **Correctness** on $\mathcal{V}$ at a given training iteration during the curriculum learning process.

## 5.5 Ablation Study

### 5.5.1 Flexibility and Compatibility

We apply SEVAL to other pseudo-label based SSL algorithms including Mean-Teacher, MixMatch and ReMixMatch and report the results with the setting of CIFAR-100 $n_1 = 50$ in Fig. 6(a). We find SEVAl can bring substantial improvements to these methods and is more effective than DASO. Of note the results of ReMixMatch w/SEVAL is higher than the results of FixMatch w/ SEVAL in Table 2 (86.7 vs 85.3). This may indicates that ReMixMatch is fit imbalanced SSL better. Due to its simplicity, SEVAL can be readily combined with other SSL algorithms that focus on LTL instead of PLR and THA. For example, SEVAL pairs effectively with the semantic alignment regularization introduced by DASO. By incorporating this loss into our FixMatch experiments with SEVAL, we were able to boost the test accuracy from 51.4 to 52.4 using the CIFAR-100 $n_1 = 50$ configuration.

We compare with the post-hoc adjustment process with LA in Fig. 6(b). We find that those post-hoc parameters can improve the model performance in the setting of CIFAR-10. In other cases, our post-hoc adjustment doesn't lead to a decrease in prediction accuracy. However, LA sometimes does, as seen in the case of STL-10. This likely stems from the complexity of the confusion matrix in such cases, where simple offsets fail to sufficiently address class bias.

We should acknowledge that for each setting, we only conduct experiments with fixed hyper-parameters (i.e., learning rate and batch size). This may be a limitation, as different hyper-parameters could lead to different results (Sohn et al., 2020). We summarize all hyper-parameters in detail in Appendix Section G, with the hope that others can build upon our work or extend the experiments to other settings.

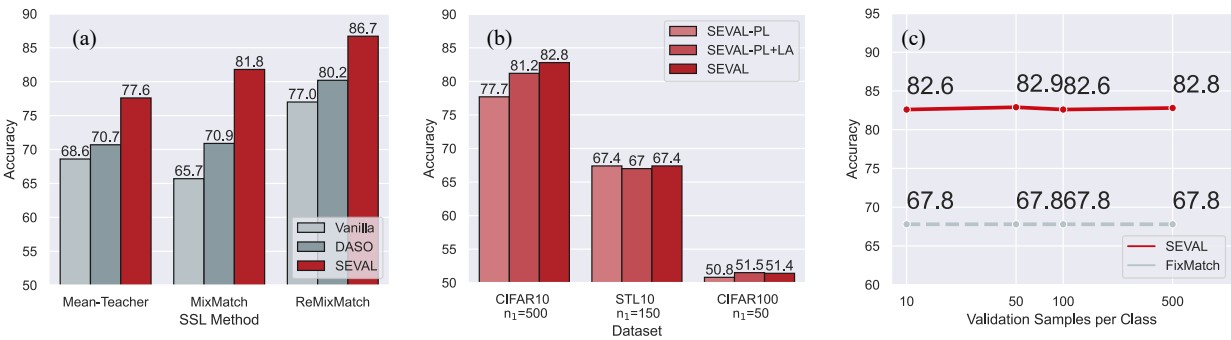

Figure 6: (a) Test accuracy when SEVAL is adapted to pseudo-label based SSL algorithms other than FixMatch under the setting of CIFAR10-LT $n_1 = 1500$. SEVAL can readily improve the performance of other SSL algorithsm. (b) Test accuracy when SEVAL employs varied types of post-hoc adjustment parameters. The learned post-hoc parameters consistently enhance performance, particularly in CIFAR-10 experiments. (c) Test accuracy when SEVAL is optimized using different validation samples under the setting of CIFAR-10 $n_1 = 500$. SEVAL requires few validation samples to learn the optimal curriculum of parameters.

### 5.5.2 Data-Efficiency

Here we explore if SEVAL requires a substantial number of validation samples for curriculum learning. To do so, we keep the training dataset the same and optimize SEVAL parameters using balanced validation dataset with varied numbers of labelled samples using the CIFAR-10 $n_1 = 500$ configuration, as shown in Fig. 6(c). We find that SEVAL consistently identifies similar $\boldsymbol{\pi}$ and $\boldsymbol{\tau}$. When we train the model using these curricula, there aren't significant differences even when the validation samples per class ranges from 10 to 500. This suggests that SEVAL is both data-efficient and resilient.

### 5.6 Analysis of Learned Thresholds

We try to determine the effectiveness of thresholds by looking into **Precision** of different classes, which should serve as approximate indicators of suitable thresholds. We illustrate an example of optimized thresholds and the learning status of FixMatch on CIFAR10-LT $n_1 = 500$ in Fig. 7(a), where SEVAL learns $\tau_c$ to be low for classes that have high **Precision**. In contrast, maximum class probability $P'_c$, does not show clear correction with **Precision**. Specifically, as highlighted with the red arrows, $P_c$ remains high for classes that exhibit high **Precision**. Consequently, maximum class probability-based threshold methods such as FlexMatch will tune the threshold to be high for classes with large $P_c$, inadequately addressing *Case 3* and *Case 4* as elaborated in Section 3.

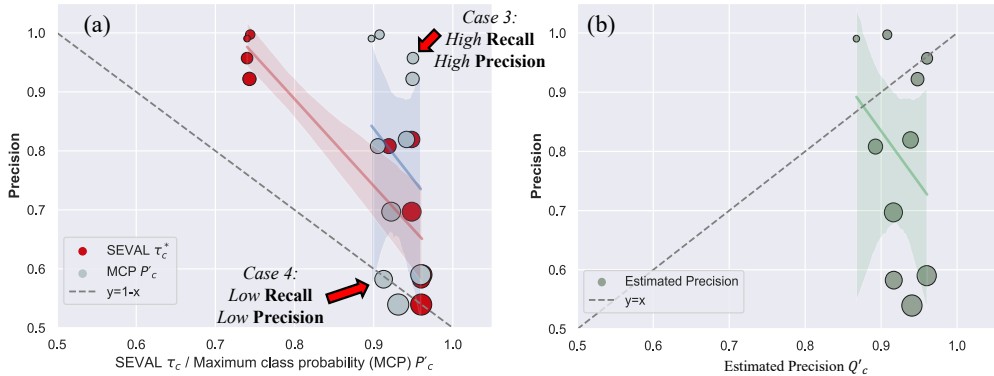

Figure 7: The correlation of different metrics between test **Precision** of FixMatch on CIFAR10-LT $n_1 = 500$. (a) The correlation of SEVAL learned $\tau_c$ and maximum class probability $P'_c$ between test **Precision**. Each point represents a class c and the size of the points indicate the number of samples in the labelled training dataset $n_c$. Note that maximum class probability $P'_c$ is the basis of current dynamic threshold method to derive thresholds. For example, FlexMatch selects more samples for classes associated with lower $P'_c$. However, as highlighted by red arrows, $P'_c$ does not correlated with **Precision** thus $P_c$ based on methods will fail *Case 3: High* **Recall** & *High* **Precision** and *Case 4: Low* **Recall** & *Low* **Precision** in accordance with Fig. 2. (b) Due to the lack of calibration in the network output probability, the estimated precision derived from the probability does not align with the actual **Precision**, thus cannot be a reliable metric to directly derive thresholds.

Instead of depending on an independent labelleddataset, we also attempt to estimate **Precision** using the model probability, so as to leverage the estimated precision to determine the appropriate thresholds. Specifically, we estimate the **Precision** of class $c$ as:

$$Q'_c = \frac{\sum_{i=1}^{K} \mathbb{1}_{ic} \max_j p_{ij}^{\mathcal{U}}}{\sum_{i=1}^{K} p_{ic}^{\mathcal{U}}}. \tag{19}$$

We visualize the estimated precision in Fig. 7(b). We find the the estimated precision does not align with the actual **Precision**. This is because the model is uncalibrated and the $Q'_c$ is heavily decided by the true positives parts (e.g. numerator in Eq. 19), thus cannot reflect the real model precision. Thus it is a essential to utilize a holdout labelled dataset to derive the optimal thresholds.

Finally, we look into the class-wise performance of SEVAL and its counterparts in Fig. 8. When compared with alternative methods, SEVAL achieves overall better performance with higher **Recall** on minority classes and higher **Precision** on majority classes. In this case, class 6 falls into *Case 3: High* **Recall** & *High* **Precision** while class 5 falls into *Case 4: Low* **Recall** & *Low* **Precision**. SEVAL shows advantages on these classes.

## 6 Conclusion and Future Work

In this study, we provide a theoretical analysis of pseudo-labeling strategies and introduce SEVAL, grounded in statistical principles, demonstrating its benefits for imbalanced SSL across diverse application scenarios. SEVAL sheds new light on pseudo-label generalization, which is a foundation for many leading SSL algorithms. SEVAL is

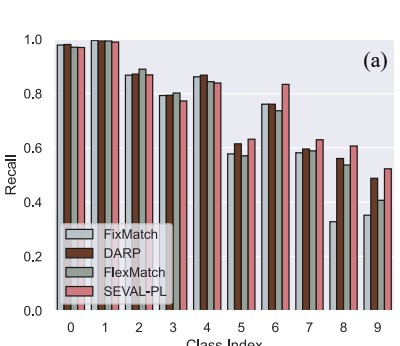 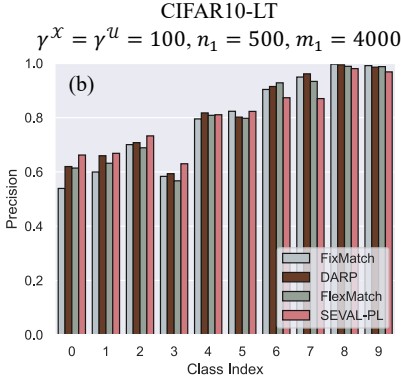 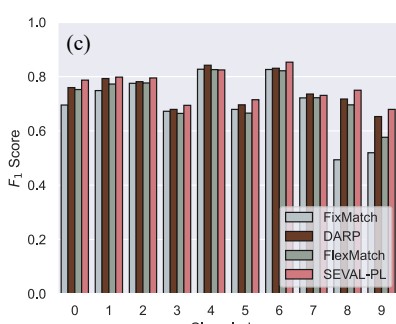

Figure 8: Class-wise performance for different SSL methods. Class indexes are arranged in descending order according to their class frequencies. We find that SEVAL achieve better overall performance than its counterparts by making neural networks more sensitive to minority classes.

both straightforward and potent, requiring no extra computation once the curriculum is acquired. As such, it can be effortlessly integrated into other SSL algorithms and paired with LTL methods to address class imbalance.

We believe that the concept of optimizing parameters or accessing unbiased learning status using a partition of the labeled training dataset could spark further innovations in long-tailed recognition and SSL. For example, our PLR optimization could be adapted to learn class-specific margins (i.e., LA during training), as discussed in LDAM (Cao et al., 2019) and other approaches (Li et al., 2020; Menon et al., 2020). This adaptation would enhance neural network learning under class imbalance and consequently benefit imbalanced SSL by yielding less biased classifiers.

We feel that the specific interplay between label refinement and threshold adjustment remains an intriguing question for subsequent research. In practice, we believe a more sophisticated curriculum incorporating a warm-up phase without unlabeled samples in early stages could further improve performance (Cascante-Bonilla et al., 2021). However, we consistently use baseline methods throughout our experiments to ensure clear interpretation of results.

In the future, by leveraging Bayesian or bootstrap techniques, we may eliminate the need for internal validation in SEVAL by improving model calibration (Loh et al., 2022; Vucetic & Obradovic, 2001). We also plan to analyze SEVAL within the theoretical framework of SSL (Mey & Loog, 2022) to acquire deeper insights. Moreover, we aim to investigate whether our findings generalize to pseudo-labels from pre-trained foundation models (Wang et al., 2022b) or are compatible with external supervision signals, such as semantic information from label names (Zhang et al., 2025).

### Acknowledgments

SJ is supported by a Wellcome Senior Research Fellowship (221933/Z/20/Z) and a Wellcome Collaborative Award (215573/Z/19/Z). The Oxford Centre for Integrative Neuroimaging is supported by core funding from the Wellcome Trust (203139/Z/16/Z). The computational aspects of this research were partly carried out at Oxford Biomedical Research Computing (BMRC), which is funded by the NIHR Oxford BRC with additional support from the Wellcome Trust Core Award Grant Number 203141/Z/16/Z.

# Appendix

## Table of Contents

## A    Glossary of Notations

| Notation | Description | Section |
|---|---|---|
| $f(\cdot)$ | A classification model. | § 1 |
| $\sigma(\cdot)$ | Softmax calculation. | § 3.1 |
| $C$ | Number of classes. | § 3.1 |
| $\boldsymbol{n} \in \mathbb{R}^C$ | Number of samples per class of $\mathcal{X}$. | § 3.1 |
| $\gamma$ | Class imbalance ratio. | § 3.2 |
| $\rho$ | Noise rate of pseudo-labels. | § 3.3 |
| $\boldsymbol{k} \in \mathbb{R}^C$ | Number of samples per class of $\mathcal{V}$. | § 4.1 |
| $\boldsymbol{\omega}^{\mathcal{V}} \in \mathbb{R}^C$ | Class weights as the reciprocal of $\boldsymbol{k}$. | § 4.2,  G |
| $X \times Y$ | Input and label space. | § 3.1 |
| $P(Y\|X)$ | Conditional probability distribution. | § 3.1 |
| $P(X, Y)$ | Joint Probability Distribution. | § 3.2 |
| $P(Y)$ | Marginal Probability Distribution. | § 3.2 |
| $\mathcal{X} = \{(\boldsymbol{x}_i, y_i)\}_{i=1}^N$ | Labelled training dataset. | § 3.1 |
| $\hat{\mathcal{U}} = \{(\boldsymbol{u}_i, \hat{y}_i)\}_{i=1}^M$ | Unlabelled training data and its associated pseudo-labels. | § 3.1 |
| $\mathcal{U} = \{(\boldsymbol{u}_i, y_i)\}_{i=1}^M$ | Unlabelled training data alongside its oracle labels. | § 3.3 |
| $\mathcal{T}$ | Test data. | § 3.2 |
| $\mathcal{V}$ | An independent labelled data, split from $\mathcal{X}$. | § 3.2 |
| $\mathcal{L}$ | Supervised loss (e.g., cross-entropy). | § 3.1 |
| $R_{\mathcal{L},\mathcal{X}}(f), \hat{R}_{\mathcal{L},\hat{\mathcal{U}}}(f)$ | Empirical risk computed on labeled and unlabeled data. | § 3.1 |
| $\mathcal{O}_w(\cdot), \mathcal{O}_s(\cdot)$ | Weak and strong operations for data augmentation. | § 3.1 |
| $\boldsymbol{p}_i \in \mathbb{R}^C$ | Model predicted probabilities. | § 3.1 |
| $\boldsymbol{q}_i \in \mathbb{R}^C$ | Pseudo-label probability. | § 3.1 |
| $\boldsymbol{\tau} \in \mathbb{R}^C$ | Class-specific filtering thresholds for pseudo-labels. | § 3.1,  4.2 |
| $\hat{\boldsymbol{z}}_i \in \mathbb{R}^C$ | Pseudo-label logit used to derive $\boldsymbol{q}_i$. | § 3.2 |
| $\boldsymbol{\pi} \in \mathbb{R}^C$ | Offsets for sample selection based on $\hat{\boldsymbol{z}}_i^{\mathcal{U}}$. | § 3.2,  4.1 |
| $t$ | The expected accuracy of pseudo-labels. | § 4.2 |
| $L$ | The length of curriculum. | § 4.3 |
| $T$ | Total training iterations. | § 4.3 |
| $\eta_\pi, \eta_\tau$ | Momentum decay ratio of offsets and thresholds. | § 4.3 |
| $\lambda$ | The Lagrange multiplier. | § 4.2 |
| $\mathcal{A}(\tau_c, c)$ | **Precision** for class $c$ when selecting samples with threshold $\tau_c$. | § 4.2 |
| $\mathcal{S}(\tau_c, c)$ | Number of class $c$ samples selected by $\tau_c$. | § 4.2,  G |
| $P'_c$ | Maximum class probability of class $c$, estimating **Recall**. | § 3.3 |
| $Q'_c$ | The estimation of **Precision** for class $c$. | § 5.6 |
| $\mathbb{1}_{ic}$ | Whether sample $i$'s most probable class is $c$. | § 3.3,  4.2 |
| $\alpha_c$ | Accuracy of samples predicted as class $c$. | § 4.2 |
| $K_c$ | (Normalized) number of samples predicted as $c$. | § 3.3,  G |
| $B$ | The size of the class group. | § G |
| $\tilde{\boldsymbol{\tau}} \in \mathbb{R}^{\lceil C/B \rceil}$ | Group-specific filtering thresholds for pseudo-labels. | § G |
| $\tilde{\mathcal{S}}(\tilde{\tau}_b, b)$ | Number of group $b$ samples selected by $\tau_b$. | § G |
| $\tilde{\alpha}_b$ | Accuracy of samples predicted within group $b$. | § G |
| $\tilde{K}_b$ | (Normalized) number of samples predicted within group $c$. | § G |

# B   Algorithms

---

**Algorithm 1** Imbalanced semi-supervised learning with SEVAL.

---

**Require:**
1: $\mathcal{X} = \{(\boldsymbol{x}_i, y_i)\}_{i=1}^N$: labelled training data, $\{\boldsymbol{u}_i\}_{i=1}^M$: unlabelled training data, $f(\cdot)$: network for classification.
2: $t$: Requested per class accuracy of the pseudo-label, $\eta_\pi$, $\eta_\tau$: Momentum decay ratio of offsets and thresholds.
3: $T$: Total training iterations, $C$: Number of classes, $L$: length of the curriculum.
4: Initialize the SEVAL parameters as $l = 1$, $\boldsymbol{\pi}^{(l)} = \underbrace{\left[1, 1, \ldots, 1\right]}_{C}$ and $\boldsymbol{\tau}^{(l)} = \underbrace{\left[0.95, 0.95, \ldots, 0.95\right]}_{C}$.

   ▷ *Estimate a curriculum of the SEVAL parameters based on a partition of the training dataset.*
5: Randomly partition $\mathcal{X}$ into two subsets, $\mathcal{X}' = \{(\boldsymbol{x}_i, y_i)\}_{i=1}^K$ and $\mathcal{V}' = \{(\boldsymbol{x}_i, y_i)\}_{i=1}^K$, each containing an equal number of data points.
6: **for iter** in $[1, \ldots, T]$ **do**
7:    Calculate the pseudo-label logit for unlabelled data $\mathcal{U}$ and obtain $\{\hat{\boldsymbol{z}}_i^{\mathcal{U}}\}_{i=1}^M$.
      ▷ *Note: FixMatch achieves this by utilizing two augmented versions of the unlabelled data.*
8:    Calculate the pseudo-label probability $\boldsymbol{q}_i = \sigma(\hat{\boldsymbol{z}}_i^{\mathcal{U}} - \log \boldsymbol{\pi}^{(l)})$.
9:    For pseudo-label $\hat{y}_i = \arg\max_j q_{ij}$ and predicted class $y_i' = \arg\max_j p_{ij}^{\mathcal{U}}$, calculate the unlabelled loss $\hat{R}_{\mathcal{L}, \hat{\mathcal{U}}}(f) = \frac{1}{M} \sum_{i=1}^M \mathbb{1}(\max_j(q_{ij}) \geq \tau_{y_i'}^{(l)}) \mathcal{L}(\hat{y}_i, \boldsymbol{p}_i^{\mathcal{U}})$.
10:   Update the network $f$ with labelled loss $\hat{R}_{\mathcal{L}, \mathcal{X}}(f)$ calculated using $\mathcal{X}'$ and $\hat{R}_{\mathcal{L}, \hat{\mathcal{U}}}(f)$ via SGD optimizer.
11:   **if** $\text{iter} \% (T/L) = 0$ **then**
12:       $l = \textbf{iter} L/T$
13:       Using the exponential moving average version of the network $f$, compute predictions on $\mathcal{V}'$ and obtain $\{\boldsymbol{z}_i^{\mathcal{V}}\}_{i=1}^K$.
14:       $\boldsymbol{\pi}^{(l)*}, \boldsymbol{\tau}^{(l)*} = \text{ESTIM}(\mathcal{V}', \{\boldsymbol{z}_i^{\mathcal{V}}\}_{i=1}^K, t)$   ▷ *SEVAL parameter estimation process.*
15:       $\boldsymbol{\pi}^{(l)} = \eta_\pi \boldsymbol{\pi}^{(l-1)} + (1 - \eta_\pi)\boldsymbol{\pi}^{(l)*}$, $\boldsymbol{\tau}^{(l)} = \eta_\tau \boldsymbol{\tau}^{(l-1)} + (1 - \eta_\tau)\boldsymbol{\tau}^{(l)*}$
16:   **end if**
17: **end for**
   ▷ *Standard SSL process.*
18: **for iter** in $[1, \ldots, T]$ **do**
19:   $l = \lceil \textbf{iter} L/T \rceil$
20:   Calculate the pseudo-label logit for unlabelled data $\mathcal{U}$ and obtain $\{\hat{\boldsymbol{z}}_i^{\mathcal{U}}\}_{i=1}^M$.
21:   Calculate the pseudo-label probability $\boldsymbol{q}_i = \sigma(\hat{\boldsymbol{z}}_i^{\mathcal{U}} - \log \boldsymbol{\pi}^{(l)})$.
22:   Calculate the unlabelled loss $\hat{R}_{\mathcal{L}, \hat{\mathcal{U}}}(f) = \frac{1}{M} \sum_{i=1}^M \mathbb{1}(\max_j(q_{ij}) \geq \tau_{y_i'}^{(l)}) \mathcal{L}(\hat{y}_i, \boldsymbol{p}_i^{\mathcal{U}})$.
23:   Update the network $f$ with labelled loss $\hat{R}_{\mathcal{L}, \mathcal{X}}(f)$ calculated using $\mathcal{X}$ and $\hat{R}_{\mathcal{L}, \hat{\mathcal{U}}}(f)$ via SGD optimizer.
24: **end for**
   ▷ *Post-hoc processing with final learned parameters; skipping over this step results in our SEVAL-PL ablation alternative.*
25: Given a test sample $\boldsymbol{x}_i$, the logit is adjusted from $\boldsymbol{z}_i$ to $\boldsymbol{z}_i - \log \boldsymbol{\pi}^{(L)*}$.

---

**Algorithm 2** SEVAL parameter estimation process, $\boldsymbol{\pi}^*$, $\boldsymbol{\tau}^* \leftarrow \text{ESTIM}\left(\mathcal{V}, \{\boldsymbol{z}_i^{\mathcal{V}}\}_{i=1}^K, t\right)$

---

**Require:**
1: $\mathcal{V} = \{(\boldsymbol{x}_i, y_i)\}_{i=1}^K$: a held-out dataset, $\{\boldsymbol{z}_i^{\mathcal{V}}\}_{i=1}^K$: the network predicted logits for $\mathcal{V}$, $t$: Requested per class accuracy of the pseudo-label.
2: $C$: Number of classes, $\boldsymbol{k}$: Number of sample per class in $\mathcal{V}$.
3: $\boldsymbol{\pi}^* = \arg\min_{\boldsymbol{\pi}} \frac{1}{K} \sum_{i=1}^K \mathcal{L}(y_i, \sigma(\boldsymbol{z}_i^{\mathcal{V}} - \log \boldsymbol{\pi}))$
   ▷ *In practice, the parameter estimation process is achieved by bound-constrained solvers.*
4: $\boldsymbol{\omega}^{\mathcal{V}} = 1/\boldsymbol{k}$   ▷ *The minority class is assigned higher weights to prioritize class-specific accuracy.*
5: **for** $c$ in $C$ **do**
6:    Get class-wise accuracy $\alpha_c = \frac{1}{K_c} \sum_{i=1}^K \omega_{y_i}^{\mathcal{V}} \mathbb{1}_{ic} \mathbb{1}(y_i = c)$
      ▷ *For each class $c$, $\mathbb{1}_{ic} = \mathbb{1}(\arg\max_j(p_{ij}^{\mathcal{V}}) = c)$ and $K_c = \sum_{i=1}^K \omega_{y_i}^{\mathcal{V}} \mathbb{1}_{ic}$ is normalized number of samples predicted as $c$.*
7:    **if** $\alpha_c < t$ **then**
8:        $\tau_c^* = \arg\min_{\tau_c} \left| \frac{1}{\mathcal{S}(\tau_c, c)} \sum_{i=1}^K \omega_{y_i}^{\mathcal{V}} \mathbb{1}_{ic} \mathbb{1}(y_i = c) \mathbb{1}(\max_j(p_{ij}^{\mathcal{V}}) > \tau_c) - t \right|$
          ▷ $\mathcal{S}(\tau_c, c) = \sum_{i=1}^K \omega_{y_i}^{\mathcal{V}} \mathbb{1}_{ic} \mathbb{1}(\max_j(p_{ij}^{\mathcal{V}}) > \tau_c)$ *is the normalized number of samples predicted as $c$ with confidence $\geq \tau_c$.*
          ▷ *This optimization can be further stabilized by performing group-based optimization, as outlined in Eq.* 16.
9:    **else**
10:       $\tau_c^* = 0$   ▷ *The quality of the pseudo-labels is satisfactory, and we make use of all of them.*
11:   **end if**
12: **end for**

---

## C Proofs

### C.1 Proof of Proposition 1

$$
\begin{aligned}
f^{\mathcal{T}}(X) &= P^{\mathcal{T}}(Y|X) \\
&= \frac{P^{\mathcal{T}}(X|Y)P^{\mathcal{T}}(Y)}{P^{\mathcal{T}}(X)} \\
&= \frac{P^{\mathcal{T}}(X|Y)P^{\mathcal{T}}(Y)}{P^{\mathcal{T}}(X)} \cdot \frac{P^{\mathcal{X}}(X)}{P^{\mathcal{X}}(X)},
\end{aligned}
\tag{20}
$$

when we assume there does not exist conditional shifts between training and test dataset following (Saerens et al., 2002), e.g. $P^{\mathcal{X}}(X|Y) = P^{\mathcal{T}}(X|Y)$, we can rewrite Eq. 20 as:

$$
\frac{P^{\mathcal{X}}(X|Y)P^{\mathcal{T}}(Y)}{P^{\mathcal{T}}(X)} \cdot \frac{P^{\mathcal{X}}(X)}{P^{\mathcal{X}}(X)}.
\tag{21}
$$

Since there is no covariate shift between the training and test domains, the relative density of $X$ is preserved across domains (i.e., $P^{\mathcal{T}}(X) \propto P^{\mathcal{X}}(X)$). Therefore, Eq. 21 can be rewritten as:

$$
\begin{aligned}
&\frac{P^{\mathcal{X}}(X|Y)P^{\mathcal{T}}(Y)}{P^{\mathcal{X}}(X)} \cdot \frac{P^{\mathcal{X}}(X)}{P^{\mathcal{T}}(X)} \\
&\propto \frac{P^{\mathcal{X}}(X|Y)P^{\mathcal{T}}(Y)}{P^{\mathcal{X}}(X)} \\
&= \frac{P^{\mathcal{X}}(X,Y)P^{\mathcal{T}}(Y)}{P^{\mathcal{X}}(X)P^{\mathcal{X}}(Y)} \\
&= \frac{P^{\mathcal{X}}(Y|X)P^{\mathcal{T}}(Y)}{P^{\mathcal{X}}(Y)} \\
&= \frac{f^{*}(X)P^{\mathcal{T}}(Y)}{P^{\mathcal{X}}(Y)}.
\end{aligned}
\tag{22}
$$

∎

### C.2 Proof of Corollary 1

$$
\begin{aligned}
&P^{\mathcal{T}}(X,Y) \\
&= P^{\mathcal{T}}(X|Y)P^{\mathcal{T}}(Y).
\end{aligned}
\tag{23}
$$

If we assume the test distribution $\mathcal{T}$ shares identical class conditionals with the unlabeled training dataset $\mathcal{U}$ (i.e. $P^{\mathcal{T}}(X|Y) = P^{\mathcal{U}}(X|Y)$), Eq. 23 can be rewritten as:

$$
\begin{aligned}
&P^{\mathcal{T}}(X|Y)P^{\mathcal{T}}(Y) \\
&= P^{\mathcal{U}}(X|Y)P^{\mathcal{T}}(Y) \\
&= \frac{P^{\mathcal{U}}(X,Y)P^{\mathcal{T}}(Y)}{P^{\mathcal{U}}(Y)}.
\end{aligned}
\tag{24}
$$

Therefore, a Bayes classifier $f^{\mathcal{T}}$ that is optimal on $P^{\mathcal{T}}(X,Y)$ should also be optimal for the resampled unlabeled dataset with distribution $P^{\mathcal{U}}(X,Y)P^{\mathcal{T}}(Y)/P^{\mathcal{U}}(Y)$.

∎

### C.3 Proof of Theorem 1

We derive the generalization bound based on the Rademacher complexity method (Bartlett & Mendelson, 2002) following the analysis for noisy label training (Natarajan et al., 2013; Liu & Tao, 2015).

**Lemma 1** *We define:*

$$\hat{\mathcal{L}}(\cdot, y) := \frac{(1-\rho)\mathcal{L}(\cdot, y) - \rho\mathcal{L}(\cdot, 3-y)}{1-2\rho}. \tag{25}$$

*With this loss function, we should have:*

$$\mathbb{E}_{\hat{y}}[\hat{\mathcal{L}}(\cdot, \hat{y})] = \mathcal{L}(\cdot, y). \tag{26}$$

Given $\hat{R}_{\hat{\mathcal{L}},\hat{\mathcal{U}}}(f) := \frac{1}{M}\sum_{i=1}^{M}\hat{\mathcal{L}}(f(\boldsymbol{u}_i), \hat{y}_i)$ the empirical risk on $\hat{\mathcal{U}}$ and $R_{\hat{\mathcal{L}},\hat{\mathcal{U}}}(f)$ is the corresponding expected risk, we have the basic generalization bound as:

$$\max_{f\in\mathcal{F}}|\hat{R}_{\hat{\mathcal{L}},\hat{\mathcal{U}}}(f) - R_{\hat{\mathcal{L}},\hat{\mathcal{U}}}(f)| \leq 2\Re(\hat{\mathcal{L}}\circ\mathcal{F}) + \sqrt{\frac{\log(1/\delta)}{2M}}, \tag{27}$$

where:

$$\Re(\hat{\mathcal{L}}\circ\mathcal{F}) := \mathbb{E}_{\boldsymbol{u}_i, \hat{y}_i, \epsilon_i}\left[\sup_{f\in\mathcal{F}}\frac{1}{M}\sum_{i=1}^{M}\epsilon_i\hat{\mathcal{L}}(f(\boldsymbol{u}_i), \hat{y}_i)\right]. \tag{28}$$

If $\mathcal{L}$ is $L$-Lipschitz then $\hat{\mathcal{L}}$ is $L_\rho$ Lipschitz with:

$$L_\rho = \frac{L}{1-2\rho}. \tag{29}$$

Based on Talagrand's Lemma (Mohri et al., 2018), we have:

$$\Re(\hat{\mathcal{L}}\circ\mathcal{F}) \leq L_\rho\Re(\mathcal{F}), \tag{30}$$

where $\Re(\mathcal{F}) := \mathbb{E}_{\boldsymbol{x}_i, \epsilon_i}\left[\sup_{f\in\mathcal{F}}\frac{1}{M}\sum_{i=1}^{M}\epsilon_i f(\boldsymbol{u}_i)\right]$ is the Rademacher complexity for function class $\mathcal{F}$ and $\epsilon_1, \ldots, \epsilon_M$ are i.i.d. Rademacher variables.

Let $\hat{f}$ be the model after optimizing with $\hat{\mathcal{U}}$, and let $f^*$ be the minimization of the expected risk $R_{\mathcal{L},\mathcal{U}}$ over $\mathcal{F}$. Then, we have:

$$
\begin{aligned}
&R_{\mathcal{L},\mathcal{U}}(\hat{f}) - R_{\mathcal{L},\mathcal{U}}(f^*) \\
&= R_{\hat{\mathcal{L}},\hat{\mathcal{U}}}(\hat{f}) - R_{\hat{\mathcal{L}},\hat{\mathcal{U}}}(f^*) \\
&= \left(\hat{R}_{\hat{\mathcal{L}},\hat{\mathcal{U}}}(\hat{f}) - \hat{R}_{\hat{\mathcal{L}},\hat{\mathcal{U}}}(f^*)\right) + \\
&\quad \left(\hat{R}_{\hat{\mathcal{L}},\hat{\mathcal{U}}}(f^*) - R_{\hat{\mathcal{L}},\hat{\mathcal{U}}}(f^*)\right) + \left(R_{\hat{\mathcal{L}},\hat{\mathcal{U}}}(\hat{f}) - \hat{R}_{\hat{\mathcal{L}},\hat{\mathcal{U}}}(\hat{f})\right) \\
&\leq 0 + 2\max_{f\in\mathcal{F}}|\hat{R}_{\hat{\mathcal{L}},\hat{\mathcal{U}}}(f) - R_{\hat{\mathcal{L}},\hat{\mathcal{U}}}(f)|
\end{aligned}
\tag{31}
$$

After we combine Eq. 27 and Eq. 31, we obtain the bound described in Theorem 1.

∎

## C.4 Proof of Lemma 1

Based on Eq. 26, we calculate the cases for $y = 1$ and $y = 2$ and obtain:

$$(1-\rho)\hat{\mathcal{L}}(\cdot, 1) + \rho\hat{\mathcal{L}}(\cdot, 2) = \mathcal{L}(\cdot, 1), \tag{32}$$

$$(1-\rho)\hat{\mathcal{L}}(\cdot, 2) + \rho\hat{\mathcal{L}}(\cdot, 1) = \mathcal{L}(\cdot, 2). \tag{33}$$

By solving the two equations, we yield:

$$\hat{\mathcal{L}}(\cdot, 1) = \frac{(1-\rho)\mathcal{L}(\cdot, 1) - \rho\mathcal{L}(\cdot, 2)}{1-2\rho}, \tag{34}$$

$$\hat{\mathcal{L}}(\cdot, 2) = \frac{(1-\rho)\mathcal{L}(\cdot, 2) - \rho\mathcal{L}(\cdot, 1)}{1-2\rho}. \tag{35}$$

∎

# D   Connections to the Neyman-Pearson Lemma

The NPL (Neyman & Pearson, 1933) provides the theoretical foundation for our THA optimization in Remark 3, where the likelihood ratio test optimally maximizes statistical power (i.e. true positive rate) at fixed significance levels (i.e. false positive rate). Our framework generalizes this principle to multi-class settings by maximizing a correctness metric (i.e. $\mathcal{A}$) under resource constraints (i.e. $\mathcal{S}$).

The NPL provides optimal threshold selection for binary decisions through constrained power maximization. Similarly, our method determines optimal multi-class thresholds by balancing two competing factors: per-class accuracy $\mathcal{A}(\tau_c, c)$ against a global prediction budget constraint $\sum_{c=1}^{C} \mathcal{S}(\tau_c, c) = M$. While the NPL relies on likelihood ratios that require known distributions, our approach operates purely through empirical accuracies, making it suitable for data-driven applications where likelihood functions are unavailable.

Through a simple example, here we demonstrate how threshold dynamics govern the optimality of balanced class accuracies. When all classes reach equal accuracy $\mathcal{A}^*$ at their optimal thresholds $\tau_c^*$, any deviation from these thresholds disrupts equilibrium. Introducing samples below $\tau_c^*$ (with accuracy $\leq \mathcal{A}^*$) reduces the average accuracy, while excluding samples above $\tau_c^*$ (with accuracy $\geq \mathcal{A}^*$) unnecessarily discards correct predictions.

This leads to the key insight: the only stable solution that simultaneously maximizes total correctness and respects the prediction budget occurs when all classes share identical accuracy $\mathcal{A}^*$. At this point, no reallocation of samples between classes can further improve the objective. This equilibrium condition represents a natural generalization of the NPL's optimality criterion, adapted for multi-class prediction with empirical thresholds.

# E   A Note on Non-Uniform Test Class Distributions

In the main paper, we assume that $P^{\mathcal{T}}(Y)$ is uniformly distributed, which represents the most common case of interest. We now discuss how the proposed method can be extended to accommodate alternative assumptions on the test distribution.

The threshold adjustment component of SEVAL operates without any assumptions regarding the test data distribution, making it directly applicable to all settings. This makes it especially suitable for imbalanced learning, where the criterion naturally assigns lower decision thresholds to minority classes, implicitly regularizing the unlabeled training data.

We now turn to the first component of SEVAL: pseudo-label refinement. Recall that in the analysis of Corollary 1, we established that the Bayes classifier learned on the test data $\mathcal{T}$, denoted $f^{\mathcal{T}}(X)$, is optimal with respect to the resampled unlabeled distribution $\frac{P^{\mathcal{U}}(X,Y)\,P^{\mathcal{T}}(Y)}{P^{\mathcal{U}}(Y)}$. Under the assumption that $P^{\mathcal{T}}(Y)$ is uniform, the unlabeled data distribution reduces to a normalization by $P^{\mathcal{U}}(Y)$. In practice, we achieve this normalization by optimizing the refinement parameter $\boldsymbol{\pi}^*$ on a held-out validation set via the following objective, which minimizes the class-balanced average cross-entropy loss:

$$\boldsymbol{\pi}^* = \arg\min_{\boldsymbol{\pi}} \sum_{j=1}^{C} \frac{1}{Ck_j} \sum_{i=1}^{K} \mathbb{1}(y_i = j)\, \mathcal{L}\Big(y_i,\, \sigma\big(\boldsymbol{z}_i^{\mathcal{V}} - \log \boldsymbol{\pi}\big)\Big), \tag{36}$$

where $\boldsymbol{z}_i^{\mathcal{V}}$ denotes the logits computed on the held-out validation set, and $k_j$ is the number of held-out samples belonging to class $j$. When $P^{\mathcal{T}}(Y)$ is non-uniform, i.e., the number of test samples per class $k_j^{\mathcal{T}}$ differs across classes, Eq. 36 generalizes to:

$$\boldsymbol{\pi}^* = \arg\min_{\boldsymbol{\pi}} \sum_{j=1}^{C} \frac{k_j^{\mathcal{T}}}{Ck_j} \sum_{i=1}^{K} \mathbb{1}(y_i = j)\, \mathcal{L}\Big(y_i,\, \sigma\big(\boldsymbol{z}_i^{\mathcal{V}} - \log \boldsymbol{\pi}\big)\Big). \tag{37}$$

This formulation allows SEVAL to be adapted to arbitrary test distributions. To validate this claim, we conduct experiments on an imbalanced training setting (CIFAR-10-LT with $n_1 = 500$). We report both per-class accuracy and several weighted averages of accuracy, each reflecting performance under a different hypothetical test distribution. Concretely, the standard average accuracy (AVG) is equivalent to the results reported in the main text. The weighted average (Weighted AVG) weights classes by their frequency in the labeled training data. We additionally consider a scenario in which minority-class samples are more likely to be drawn at test time (Inverse Weighted AVG). By incorporating the target test class frequencies $k_j^{\mathcal{T}}$ into the training objective via Eq. 37, we obtain three model variants: SEVAL, SEVAL-W, and SEVAL-INV, respectively. Results are reported in Table 7. Each variant achieves the best performance under its corresponding test condition. When the test distribution matches the training distribution,

the adjustment yields only marginal gains, as the model already exhibits the appropriate class bias; however, in the other two scenarios, the improvements are more pronounced.

Table 7: Class-wise test accuracy analysis on CIFAR10-LT with $n_1 = 500$. We present per-class accuracy together with several weighted averages of class-wise accuracy, capturing evaluation scenarios where the test data follows different class distributions.

| Algorithm | Class | | | | | | | | | | AVG (Reported) | Weighted AVG | Inverse Weighted AVG |
|---|---|---|---|---|---|---|---|---|---|---|---|---|---|
| | c=1 (40.5%) | c=2 (24.2%) | c=3 (14.5%) | c=4 (8.7%) | c=5 (5.2%) | c=6 (3.1%) | c=7 (1.9%) | c=8 (1.1%) | c=9 (0.6%) | c=10 (0.4%) | | | |
| FixMatch | 98.8 | 99.5 | 88.1 | 80.6 | 85.7 | 59.0 | 76.6 | 60.1 | 35.5 | 28.4 | 71.2 | 92.4 | 44.6 |
| w/ DARP | 97.6 | 99.6 | 89.2 | 79.6 | 87.0 | 60.3 | 76.8 | 57.7 | 57.1 | 42.1 | 74.7 | 92.3 | 55.1 |
| w/ FlexMatch | 96.4 | 99.6 | 89.3 | 79.5 | 86.2 | 61.7 | 77.0 | 60.8 | 48.2 | 38.2 | 73.7 | 91.8 | 51.9 |
| w/ SEVAL | 89.3 | 97.3 | 81.4 | 65.5 | 85.4 | 79.7 | 85.1 | 67.7 | 89.2 | 84.5 | **82.5** | 87.2 | 82.8 |
| w/ SEVAL-W | 98.0 | 99.5 | 87.6 | 78.7 | 87.9 | 65.5 | 76.1 | 60.6 | 66.8 | 59.5 | 78.0 | **92.5** | 65.1 |
| w/ SEVAL-INV | 81.1 | 93.1 | 71.2 | 55.2 | 77.5 | 75.0 | 89.9 | 83.4 | 92.0 | 86.3 | 80.5 | 80.2 | **86.0** |

# F    Additional Experiments

In this section, we present additional experimental results conducted under various settings to assess the generalizability of SEVAL.

## F.1    Sensitivity Analysis

Table 8: Sensitivity analysis of hyper-parameters $t$, $\eta_\pi$ and $\eta_\tau$ across experimental settings. Best results are in **bold** for each configuration.

| CIFAR10-LT, $\gamma^{\mathcal{X}} = \gamma^{\mathcal{U}} = 100$ $n_1 = 500$, $m_1 = 4000$ | | CIFAR10-LT, $\gamma^{\mathcal{X}} = \gamma^{\mathcal{U}} = 100$ $n_1 = 1500$, $m_1 = 3000$ | |
|---|---|---|---|
| $t = 0.6$ | $82.5 \pm 0.45$ | $t = 0.6$ | **85.6** $\pm 0.24$ |
| $t = 0.7$ | $82.2 \pm 0.11$ | $t = 0.7$ | $85.3 \pm 0.27$ |
| $t = 0.75$ (reported) | **82.8** $\pm 0.56$ | $t = 0.75$ (reported) | $85.3 \pm 0.25$ |
| $\eta_\pi = 0.995$ | $81.4 \pm 0.36$ | | |
| $\eta_\pi = 0.999$ (reported) | **82.8** $\pm 0.56$ | CIFAR100-LT, $\gamma^{\mathcal{X}} = \gamma^{\mathcal{U}} = 10$ $n_1 = 50$, $m_1 = 400$ | |
| $\eta_\pi = 0.9995$ | $82.5 \pm 0.35$ | | |
| $\eta_\tau = 0.995$ | $81.5 \pm 0.38$ | $t = 0.4$ | $50.9 \pm 0.22$ |
| $\eta_\tau = 0.999$ (reported) | $82.8 \pm 0.56$ | $t = 0.5$ (reported) | **51.4** $\pm 0.95$ |
| $\eta_\tau = 0.9995$ | **82.9** $\pm 0.09$ | $t = 0.6$ | $51.3 \pm 0.11$ |

We perform experiments with SEVAL, varying the core hyperparameters, and present the results in Table 8. Our findings indicate that SEVAL exhibits robustness, showing insensitivity to hyper-parameter variations within a reasonable range.

Class-wise accuracy $t$ serves as a replacement for the confidence threshold $\tau$ that existing SSL methods (e.g., FixMatch) employ to filter out low-confidence samples. While SEVAL remains stable across a range of $\tau$ values under typical SSL settings, performance may degrade when the unlabeled data contains out-of-distribution samples, as domain shift between the labeled and unlabeled data can render the optimization process suboptimal.

## F.2    Results on ImageNet-127

We conduct experiments on Small ImageNet-127 (Su & Maji, 2021) with a reduced image size of $32 \times 32$ following (Fan et al., 2022). ImageNet-127 consolidates the 1000 classes of ImageNet into 127 categories according to the WordNet hierarchy. This results in a naturally long-tailed class distribution with an imbalance ratio of $\gamma^{\mathcal{X}} \approx \gamma^{\mathcal{U}} \approx 286$. We randomly select 10% of the training samples as the labeled set and utilize the remainder as unlabeled data. We conduct experiments using ResNet-50 and Adam optimizer. Results are summarized in Table 9.

Table 9: Averaged class recall on Small-ImageNet-127. Best results within the same category are in **bold** for each configuration. Training sample counts per class range from $n_1 = 28{,}013$ to $n_{127} = 89$.

| | Method type | | | Small- |
|---|---|---|---|---|
| Algorithm | LTL | PLR | THA | ImageNet-127 |
| FixMatch | | | | 29.4 |
| w/ FreeMatch | | ✓ | ✓ | 30.0 |
| w/ SEVAL-PL | | ✓ | ✓ | **31.0** |
| w/ SAW | ✓ | | | 29.4 |
| w/ DASO | ✓ | ✓ | | 29.4 |
| w/ SEVAL | ✓ | ✓ | ✓ | **34.8** |

### F.3 Integration with Other SSL Frameworks

As an extension to results in Fig. 6, we summarize the results when introducing SEVAL into other SSL frameworks in Table 10. We summarize the implementation details of those methods in Section G.

Table 10: Accuracy on CIFAR10-LT based on SSL methods other than FixMatch. Best results within the same category are in **bold** for each configuration.

| Algorithm | CIFAR10-LT $\gamma^{\mathcal{X}} = \gamma^{\mathcal{U}} = 100$ $n_1 = 1500$ $m_1 = 3000$ | CIFAR100-LT $\gamma^{\mathcal{X}} = \gamma^{\mathcal{U}} = 10$ $n_1 = 150$ $m_1 = 300$ |
|---|---|---|
| Mean Teacher (Tarvainen & Valpola, 2017) | 68.6 ±0.88 | 52.1 ±0.09 |
| w/ DASO (Oh et al., 2022) | 70.7 ±0.59 | 52.5 ±0.37 |
| w/ SEVAL | **77.6** ±0.63 | **53.8** ±0.24 |
| MixMatch (Berthelot et al., 2019b) | 65.7 ±0.23 | 54.2 ±0.47 |
| w/ DASO (Oh et al., 2022) | 70.9 ±1.91 | 55.6 ±0.49 |
| w/ SEVAL | **81.8** ±0.82 | **57.8** ±0.26 |
| ReMixMatch (Berthelot et al., 2019a) | 77.0 ±0.55 | 61.5 ±0.57 |
| w/ DASO (Oh et al., 2022) | 80.2 ±0.68 | 62.1 ±0.69 |
| w/ SEVAL | **86.7** ±0.71 | **63.1** ±0.38 |

## G Implementation Details

### G.1 Benchmarks

We conduct experiments upon the code base of (Oh et al., 2022) for experiments of CIFAR10-LT, CIFAR100-LT and STL10-LT. We take some baseline results from the DASO paper (Oh et al., 2022) to Table 2, Table 4 and Table 10. including the results of supervised baselines, DARP, CReST+, ABC and DASO. We implement all comparison methods using the same codebase. In SimPro (Du et al., 2024a), unlabeled training uses prior-adjusted soft pseudo-labels. However, this becomes unstable in low-label settings: CIFAR-10 with $n_1 = 500$, CIFAR-100 with $n_1 = 50$, and all STL-10 runs occasionally fail to converge. We address this by adopting FixMatch-style hard pseudo-labels in these settings while retaining SimPro's prior estimation and adaptation modules.

As DASO (Oh et al., 2022) does not supply the code for the Semi-Aves experiments, we conduct all the experiments for this setting ourselves. We train ResNet-50 (He et al., 2016) which is pretrained on ImageNet (Deng et al., 2009) for the task of Semi-Aves following (Su & Maji, 2021). In accordance with (Oh et al., 2022), we merge the training and validation datasets provided by the challenge, yielding a total of 5959 samples for training which come from 200 classes. We conduct experiments utilizing 26,640 unlabelled samples which share the same label space with $\mathcal{X}$ in the $\mathcal{U} = \mathcal{U}_{in}$ setting, and 148,848 unlabelled samples of which 122,208 are from open-set classes in the $\mathcal{U} = \mathcal{U}_{in} + \mathcal{U}_{out}$

setting. For experiments on Semi-Aves, we set the base learning rate as 0.005. We train the network for 45,000 iterations. The learning rate is linear warmed up during the first 2500 iterations, and degrade after 15,000 and 30,000, with a factor of 10. We choose training batch size as 32. The images are firstly cropped to $256 \times 256$. During training, the images are then randomly cropped to $224 \times 224$. At inference time, the images are cropped in the center with size $224 \times 224$.

## G.2    Hyper-Parameters

To facilitate reproducibility, we summarize all selected hyperparameters for our experiments in Table 11.

Table 11: Experiment-specific hyper-parameters. $t^{\mathcal{V}}$ is the required accuracy if we directly optimize $\boldsymbol{\tau}$ along the training process using a separate validation dataset.

| Hyper-parameter | CIFAR10-LT, $\gamma^{\mathcal{X}}=100$ $n_1=500$ $\quad$ $n1=1500$ $m_1=4000$ $\quad$ $m_1=3000$ | | CIFAR100-LT, $\gamma^{\mathcal{X}}=10$ $n_1=50$ $\quad$ $n_1=150$ $m_1=400$ $\quad$ $m_1=300$ | | STL10-LT, $\gamma^{\mathcal{X}}=20$ $n_1=150$ $\quad$ $n_1=450$ $M=100,000$ | | Semi-Aves $\mathcal{U}=\mathcal{U}_{\text{in}}$ $\quad$ $\mathcal{U}=\mathcal{U}_{\text{in}}+\mathcal{U}_{\text{out}}$ | |
|---|---|---|---|---|---|---|---|---|
| $C$ | 10 | | 100 | | 10 | | 200 | |
| $T$ | 250,000 | | 250,000 | | 250,000 | | 45,000 | |
| $t$ | 0.75 | | 0.5 | 0.65 | 0.7 | 0.6 | 0.9 | 0.99 |
| $t^{\mathcal{V}}$ | 0.9 | | 0.65 | 0.7 | 0.95 | 0.85 | —— | |
| $L$ | 500 | | 100 | | 500 | | 90 | |
| $B$ | 2 | | 25 | 10 | 2 | 1 | 10 | |
| $\eta_\pi$ | 0.999 | | 0.95 | 0.9 | 0.995 | | 0.99 | 0.9 |
| $\eta_\tau$ | 0.999 | | 0.95 | 0.9 | 0.9995 | 0.999 | 0.99 | 0.9 |

We find that the value of $t$ does not significantly affect performance, and SEVAL works well across a variety of reasonable choices, as shown in Table 8. For curriculum length $L$, smaller values should be chosen when the number of classes $C$ is large, since parameter optimization becomes more time-consuming with larger $C$. Additionally, the exponential momentum parameters ($\eta_\pi$ and $\eta_\tau$) should be adjusted accordingly. We also suggest using larger momentum values ($\eta_\pi$ and $\eta_\tau$) and a larger group size $B$ when training data is scarce, as this helps smooth the learned parameters along the training process.

Furthermore, in practice, we find that in imbalanced SSL settings, the minority classes sometimes have very few samples, making it difficult to optimize the thresholds correctly based on Eq. 16. In this case, we also set the learned $\tilde{\tau}_b^*$ to be 0, in order to leverage more data from the minority classes. Formally, we denote $\tilde{K}_b = \sum_{c=bB+1}^{bB+B} \sum_{i=1}^{K} \omega_{y_i}^{\mathcal{V}} \mathbb{1}_{ic}$ as the normalized number of predicted samples within group $b$. When $\tilde{K}_b < \sum_{i=1}^{K} \frac{B\omega_{y_i}^{\mathcal{V}}}{e_1 C}$ or $\sum_{i=1}^{K} \mathbb{1}(y_i = c) < e_2$, where $e_1$ and $e_2$ are hyper-parameters that we both set to 10 for all experiments, we also have $\tilde{\tau}_b^* = 0$ and keep their corresponding $\pi_c$ within group $b$ as low as $\pi_c = \min_j(\pi_j)$. This implies:

- In instances where the models exhibit a pronounced bias, limiting their capability to detect over 10% of the samples within a particular group, we adjust the associated thresholds and consequently increase our sample selection.
- When a group comprises fewer than 10 samples, the feasibility of optimizing thresholds based on proportion diminishes, necessitating an enhanced sample selection.

In Table 12, we summarize the per-class sample counts in the held-out data (denoted as $k_c$) for most experiments, providing guidance for parameter selection in other applications. In practice, we split the training dataset in half, resulting in $\boldsymbol{k} = \boldsymbol{n}/2$. Consequently, the reader can readily infer $k_c$ for other configurations, such as when a different imbalance ratio (i.e., $\gamma^{\mathcal{X}}$) or a different number of classes (i.e., $n_1$) is chosen for stress testing. The classes are listed in descending order of sample counts, with $k_1$ being the largest and $k_C$ the smallest. For ImageNet-127, we maintain a constant $k_c = 10$ across all classes, since the dataset's tail classes are relatively more populated ($n_C = 89$).

## G.3    SEVAL with Other SSL Algorithms

Here, we provide implementation details of how SEVAL can be integrated into other pseudo-labeling based SSL algorithms.

Specifically, we apply SEVAL to Mean Teacher (Tarvainen & Valpola, 2017), MixMatch (Berthelot et al., 2019b) and ReMixMatch (Berthelot et al., 2019a). These algorithms produce pseudo-label $\hat{y}_i$ based on its corresponding

Table 12: Per-class labeled sample counts in held-out datasets. $k_c(\times\text{COUNT})$ denotes $k_c$ samples per class in the held-out dataset, repeated COUNT times.

| Dataset | Per-class counts (sorted descending) |
|---|---|
| CIFAR10-LT, $\gamma^{\mathcal{X}}$=100, $n_1$=500 | 500, 297, 178, 107, 64, 38, 23, 14, 8, 5 |
| CIFAR100-LT, $\gamma^{\mathcal{X}}$=10, $n_1$=50 | 50, 48, 47, 46, 45, 44, 43, 42, 41, 40, 39, 38, 37, 36($\times$2), 35, 34, 33, 32($\times$2), 31, 30, 29($\times$2), 28($\times$2), 27($\times$2), 26($\times$2), 25, 24($\times$2), 23($\times$2), 22($\times$2), 21($\times$2), 20($\times$2), 19($\times$2), 18($\times$2), 17($\times$3), 16($\times$2), 15($\times$3), 14($\times$3), 13($\times$4), 12($\times$4), 11($\times$4), 10($\times$4), 9($\times$4), 8($\times$5), 7($\times$5), 6($\times$7), 5($\times$8), 4($\times$9), 3($\times$10), 2($\times$8) |
| STL10-LT, $\gamma^{\mathcal{X}}$=20, $n_1$=150 | 150, 107, 77, 55, 39, 28, 20, 14, 10, 7 |
| Semi-Aves | 26($\times$2), 23($\times$3), 22($\times$6), 21($\times$5), 20($\times$9), 19($\times$10), 18($\times$15), 17($\times$16), 16($\times$18), 15($\times$18), 14($\times$18), 13($\times$12), 12($\times$20), 11($\times$12), 10($\times$12), 9($\times$16), 8($\times$7), 7($\times$1) |

pseudo-label probability $q_i$ and logit $\hat{z}_i^{\mathcal{U}}$ in different ways. SEVAL can be easily adapted by refining $q_i$ using the learned offset $\pi^*$.

It should be noted that these SSL algorithms do not include the process of filtering out pseudo-labels with low confidence. Therefore, for simplicity and fair comparison, we do not include the threshold adjustment into these methods. We expect that SEVAL can further enhance performance through threshold adjustment and plan to explore this further in the future.

### G.3.1 Mean Teacher

Mean Teacher generates pseudo-label logit $\hat{z}_i^{\mathcal{U}}$ based on a EMA version of the prediction models. SEVAL calculates the pseudo-label probability as $q_i = \sigma(\hat{z}_i^{\mathcal{U}} - \log \pi^*)$, which is expected to have less bias towards the majority class.

### G.3.2 MixMatch

MixMatch calculates $\hat{y}_i$ based on multiple transformed version of an unlabelled sample $u_i$. SEVAL adjusts each one of them with $\pi^*$, separately.

### G.3.3 ReMixMatch

ReMixMatch proposes to refine pseudo-label probability $q_i$ with distribution alignment to match the marginal distributions. SEVAL adjusts the the probability using $q_i = \sigma(\hat{z}_i^{\mathcal{U}} - \log \pi^*)$ before ReMixMatch's process including distribution alignment and temperature sharpening.

## H   Class-Wise Performance

Table 13: The number of classes of different performance when trained with FixMatch. We demonstrate the prevalent occurrence of the four class performance scenarios in existing semi-supervised learning tasks.

| Model Performance | CIFAR10-LT $\gamma^{\mathcal{X}} = \gamma^{\mathcal{U}} = 100$ | | CIFAR100-LT $\gamma^{\mathcal{X}} = \gamma^{\mathcal{U}} = 10$ | | STL10-LT $\gamma^{\mathcal{X}} = 20, \gamma^{\mathcal{U}}: unknown$ | | Semi-Aves | |
|---|---|---|---|---|---|---|---|---|
| | $n_1 = 500$ $m_1 = 4000$ | $n_1 = 1500$ $m_1 = 3000$ | $n_1 = 50$ $m_1 = 400$ | $n_1 = 150$ $m_1 = 300$ | $n_1 = 150$ | $n_1 = 450$ $M = 100,000$ | $\mathcal{U} = \mathcal{U}_{\text{in}}$ | $\mathcal{U} = \mathcal{U}_{\text{in}} + \mathcal{U}_{\text{out}}$ |
| *Case 1: High* **Recall** & *Low* **Precision** | 4 | 5 | 24 | 29 | 2 | 4 | 36 | 45 |
| *Case 2: Low* **Recall** & *High* **Precision** | 4 | 4 | 18 | 20 | 2 | 2 | 35 | 45 |
| *Case 3: High* **Recall** & *High* **Precision** | 1 | 1 | 33 | 29 | 3 | 2 | 69 | 58 |
| *Case 4: Low* **Recall** & *Low* **Precision** | 1 | 0 | 25 | 22 | 3 | 2 | 60 | 52 |
| Total classes | 10 | 10 | 100 | 100 | 10 | 10 | 200 | 200 |

Here we summarize the number of classes of different performance, as demonstrated in Table. 13. We report the model performance of FixMatch trained on different datasets. Here we consider a class to have high **Recall** when its performance surpasses the average **Recall** of the classes. The same principle applies to **Precision**.

We observe that $Case1$ is frequently encountered in the majority class, while $Case2$ is commonly observed in the minority class. $Case3$ and $Case4$ also widely exist in different scenarios. Theoretically, SEVAL's threshold learning strategies better fit Case 3 and Case 4 compared to existing methods, consequently achieving superior performance.

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
