# OpenReview forum: "Imbalanced Semi-Supervised Learning via Label Refinement and Threshold Adjustment"
_TMLR — Accepted by TMLR_

### Review · Reviewer_V61Q · 2025-11-19

**Summary Of Contributions:**

The paper introduces SEVAL, which is validation-data guided framework for improving pseudo-labels in imbalanced semi-supervised learning by unifying pseudo-label refinement (PLR, logit offsets) and threshold adjustment (THA) into a statistically motivated curriculum learned from a small labeled subset that acts as a proxy for the test distribution. It introduces theory showing why existing heuristic PLR/THA methods are suboptimal, develops class-balanced and group-based estimators to handle imbalanced or tiny validation sets, and provides experiments—including CIFAR-LT, STL-LT, ImageNet-127, and Semi-Aves—that demonstrate gains and robustness to label scarcity, varying imbalance ratios, validation-set size, and hyper-parameters when combined with FixMatch.

**Additional Comments:**

I feel the paper is a bit too long and makes suboptimal choices about what is in the main paper and what in the appendix (see also above). I think it could benefit from some re-arrangements, but leave this as optional recommendation and not a requested change.

**Audience:**

Yes

**Audience Explanation:**

- Semi-supervised learning under class imbalance is a practically relevant problem, and pseudo-labelling is a widely adopted approach. Improvements to this approach are thus relevant and of interest to the TMLR community.
 - Identification of limitations of existing methods and developing a theoretical framework (Section 3) can be the base for follow-up work by the community.
 - The proposed method is somewhat incremental compared to prior work such as FlexMatch (Zhang et al., 2021) and Dash (Xu et al., 2021). I think it meets the (lower) bar in terms of novelty that TMLR has, but the paper should discuss these closely related works in more detail and make the relationship clearer.

**Broader Impact Concerns:**

no concerns, addressing bias in imbalanced semi-supervised learning should be broadly beneficial.

**Claims And Evidence:**

No

**Claims Explanation:**

- The proposed method is supported by extensive experiments on several datasets like CIFAR-LT, STL-LT, ImageNet-127, and Semi-Aves.
-  The theoretical analysis of limitations of existing methods (Section 3) provides some support for the claims of the paper.
- A major remaining issue is (as also stated by the authors) that in practical scenarios, there would be strong constraints on available validation set (such as imbalances in held-out data set itself limited sample size per class in validation data). While this paper discusses these issues in Appendix D, it remains unclear to which extent experiments presented in the main paper were using imbalanced and small validation sets. That is: the paper should always report k_c (number of sample per class in the validation dataset)

**Requested Changes:**

- The discussion on learning with imbalanced held-out data (Appendix D.1) and "few held-out data points per class" (Appendix D.2) are more than just implementation details. They are essential contributions to make the proposed method practically applicable. They should be moved to the main paper and be studied more extensively empirically.
- The literature review and related work is somewhat outdated (essentially no post-2022 works are covered). Also relevant related work like DASH are not discussed in detail. Overall, it remains unclear which methodological contributions are actually not already covered in existing literature.

---

### Review · Reviewer_oKeb · 2026-03-05

**Summary Of Contributions:**

**Summary**

This paper studies semi-supervised learning under class imbalance, focusing on the pseudo-labeling pipeline. It argues that two popular families: 1) pseudo-label refinement and 2) threshold adjustment are typically heuristic and often fall to suboptimal. The authors provide a statistical analysis showing 1) PLR should depend on the target/test label distribution rather than the unlabeled training distribution, and 2) class-wise thresholding should be driven by Precision rather than Recall, since Recall-style signals are not identifiable from unlabeled data alone. Authors then propose SEVAL, a unified framework that learns logit offsets for PLR and class-wise thresholds for THA from a held-out labeled subset. The method learns a curriculum to track how logits vary during training.

**Strengthen**
- The insight of decomposing imbalanced SSL into PLR and THA is interesting, and well-motivated.

- Experiments and ablations show clear improvement of each component, i.e., SEVAL-PLR and SEVAL-THA outperform their closest counterparts.

- Evaluation on realistic long-tailed dataset (Semi-Aves) demonstrate the effectiveness of the proposed methods on real-world problems.

**Weakness**

- The theoretical assumption includes uniform test priors, which may limit direct applicability to real-world settings where test priors are also long-tailed (the paper partly addresses this by using class-wise recall for ImageNet-127).

**Audience:**

Yes

**Audience Explanation:**

Imbalanced SSL is a common real-world pain point, and the proposed method offers several insights:
- Pseudo-label refinement should be test-prior-aware rather than based on unlabeled-prior alignment.
- Precision-driven class-wise thresholding is a more effective principle than recall-driven heuristics for selecting reliable pseudo-labels.

As a result, the paper is likely to attract attention and provide practical guidance for TMLR audience, particularly for those working on improving long-tailed SSL in practice.

**Broader Impact Concerns:**

No Broader Impact

**Claims And Evidence:**

Yes

**Claims Explanation:**

The central claims are 1) existing PLR/THA heuristics can be theoretically misaligned, and 2) learning PLR offsets and THA thresholds from held-out labeled data improves pseudo-label quality and downstream performance. The paper provides evidence including:
- Analysis of thresholding by recal and by precision.
- SEVAL and SEVAL-PL outperform DARP/FlexMatch/FreeMatch and hybrid baselines DASO/ACR across multiple benchmarks.
- Ablations showing each component improves the performance, and the combination presents optimal results.
- Analyses of training dynamics indicate their method avoids late-stage degradation seen in DARP/LA in their pseudo-label “Gain” curves.

**Requested Changes:**

Add discussion on non-uniform targets. The authors already assume uniform $P^T(Y)$ for most benchmarks and switch metrics for ImageNet-127. Please add a discussion about how SEVAL would behave if the evaluation prior is also long-tailed, and whether the proposed optimization objective can be adapted accordingly. Apart from this, I have no further requests for changes as the current paper is well-structured.

---

> ### Author Response · Authors · 2026-04-03
> **Author Response to Reviewer oKeb for Paper 6142**
>
> We thank the reviewer for taking the time to assess our manuscript and for their valuable suggestions. Based on the reviewer's feedback, we have added a discussion on the application of SEVAL to non-uniform test data distributions. Please find our revisions highlighted in $\textcolor{purple}{purple}$ in Appendix Section E. Below is our response to the comment.
>
> > **Q1:** Please add a discussion about how SEVAL would behave if the evaluation prior is also long-tailed, and whether the proposed optimization objective can be adapted accordingly.
>
> **A1:** Thank you for raising this point. Choosing a uniform test distribution is a common practice, as it best reflects the class-average accuracy we aim to achieve when training on imbalanced data; we therefore adopt it as our default setting. As the reviewer highlights, we argue that the pseudo-label refinement process should be linked to the target test distribution, and the logits should be adjusted based on $\frac{P^\mathcal{T}(Y)}{P^\mathcal{U}(Y)}$. Within our framework, the modification is straightforward: we simply reweight the per-class loss according to $P^\mathcal{T}(Y)$. We implement this on CIFAR10-LT and find that this strategy consistently outperforms baselines, especially when $P^\mathcal{T}(Y)$ differs from the training distribution. Specifically, under an inverse imbalanced test distribution, accuracy improves from 44.6 to 86.0, while gains are more modest when the test distribution matches training (92.4 to 92.5). Existing imbalanced SSL methods lack this adaptability, further supporting the value of a flexible pseudo-label refinement strategy.

---

### Review · Reviewer_uicZ · 2026-03-09

**Summary Of Contributions:**

This paper studies class imbalance in semi-supervised learning (SSL) through a statistical analysis of pseudo-label generation and selection, and proposes SEVAL. This framework learns pseudo-label refinement offsets and class-wise threshold adjustments from a class-balanced validation set. The authors show that optimal pseudo-label refinement depends on the test-class prior and argue that thresholds should be equalised per class to achieve precision. Experiments on long-tailed SSL benchmarks demonstrate consistent improvements over baselines.

**Additional Comments:**

1. How is the target precision t chosen in practice? Please provide sensitivity analyses across datasets and imbalance ratios, and discuss whether t can be adapted online.
2. How frequently are π and τ updated during training (curriculum length, EMA coefficients), and what is the computational overhead relative to the base SSL?
3. Could thresholds be made soft rather than hard 0/1 selection?
4. What happens if the test prior is not uniform?

**Audience:**

Yes

**Audience Explanation:**

- Proposes a unified framework that learns per-class logit offsets and thresholds for pseudo-label calibration under class imbalance.
- Evaluates on multiple long-tailed SSL benchmarks and shows consistent improvements over selected baselines.
- The paper clearly decomposes the problem into PLR and THA components and provides intuitive figures and descriptions of the optimisation and training curriculum.

**Broader Impact Concerns:**

- The claim that thresholds should equalise per-class precision relies on strong approximations and is closer to a heuristic than a formal guarantee. The logit-offset formulation largely parallels existing label-shift correction and calibration methods, making the core novelty somewhat incremental.
 - Some theoretical claims rest on strong approximations, and the PLR component closely aligns with known label-shift/cross-entropy calibration ideas, making the novelty more incremental than foundational.

**Claims And Evidence:**

Yes

**Claims Explanation:**

The empirical evidence is fairly clear and generally supports the main practical claim that the proposed method improves performance on several imbalanced SSL benchmarks, and the paper also provides intuitive toy examples and ablations to motivate its design.

**Requested Changes:**

- SEVAL requires a held-out labelled subset. This introduces two concerns: in SSL benchmarks, labelled data is already scarce, and splitting labelled data reduces the training set. The paper argues that the split can be small, but the trade-off is not analysed.
- Limited comparison to recent SSL methods. The paper mainly compares against older SSL approaches.
- Limited analysis of pseudo-label noise. The paper assumes that reducing pseudo-label noise improves SSL (Theorem 1), which is intuitive. However, no empirical measurement of pseudo-label noise is shown.

---

> ### Author Response · Authors · 2026-04-03
> **Author Response to Reviewer uicZ for Paper 6142 (1/3)**
>
> We thank the reviewer for their careful reading and constructive feedback, which has helped us improve the manuscript. We have addressed all comments in the revised version; the main changes are summarized below.
>
> 1. **Extended "Related Work" section**, which now includes discussions of recent SSL methods (SimPro (NeurIPS 2024), CoLA (ICLR 2026), MetaExpert (ICML 2025), and CPG (NeurIPS 2025)).
> 2. **Extended the computational efficiency discussion** in the methods section.
> 3. **Highlighted experimental details** (e.g., curriculum length $L$) in the experiments section.
> 4. **Added results and implementation of two recent comparison methods** (SimPro and MetaExpert) in the main experiments.
> 5. **Highlighted that SEVAL can be further integrated with other long-tail learning algorithms**, clarifying that analyzing improved pseudo-label refinement and threshold adjustment strategies is the primary goal of this paper.
> 6. **Enriched the discussion of sample efficiency** in the experiments section.
> 7. **Enriched the analysis of pseudo-label noise** with additional evidence across more datasets and settings in the experiments section.
> 8. **Added a discussion of online tuning of $t$**, proposing $\textbf{Correctness}$ on $\mathcal{V}$ as a practical criterion for dynamically updating $t$ throughout training.
> 9. **Added a discussion and experiments on adapting SEVAL to non-uniform test distributions**, demonstrating consistent gains over baselines under varied test priors.
> 10. **Enriched the ablation study on $t$** across multiple datasets and imbalance ratios.
>
> Please find our revisions highlighted in $\textcolor{green}{green}$ in the manuscript. Below, we provide point-by-point responses to your comments.
>
> > **Q1:** SEVAL requires a held-out labelled subset. This introduces two concerns: in SSL benchmarks, labelled data is already scarce, and splitting labelled data reduces the training set. The paper argues that the split can be small, but the trade-off is not analysed.
>
> **A1:** In the standard SSL training process, we utilize the same training data as other SSL algorithms. Prior to the standard SSL process, we split the training dataset in half and learn the curriculum of $\boldsymbol{\pi}$ and $\boldsymbol{\tau}$ used thereafter. Although reserving a held-out subset reduces the effective training set, the curriculum parameters $\boldsymbol{\pi}$ and $\boldsymbol{\tau}$ learned on this partition generalize well to the full training data, as we demonstrate across all experiments.
>
> We have further clarified this in Section 4.3 and the analysis in Section 5.3.
>
> > **Q2:** Limited comparison to recent SSL methods. The paper mainly compares against older SSL approaches.
>
> **A2:** Thank you for your suggestion. We carefully studied the recent literature and compare SEVAL against recent SSL methods (SimPro (NeurIPS 2024), CoLA (ICLR 2026), MetaExpert (ICML 2025), and CPG (NeurIPS 2025)). We find that recent studies also demonstrate the effectiveness of producing less biased pseudo-labels, but lack a principled perspective for achieving statistically optimal offsets and refinements, which is the main focus of this paper. We also implement two of them (SimPro and MetaExpert) in our codebase for a fair comparison and observe that SEVAL still outperforms these methods across all main experiments.
>
> We have updated the related works section and reported the results in Table 2.
>
> > **Q3:** Limited analysis of pseudo-label noise. The paper assumes that reducing pseudo-label noise improves SSL (Theorem 1), which is intuitive. However, no empirical measurement of pseudo-label noise is shown.
>
> **A3:** We report an analysis of pseudo-label accuracy in Section 5.4. In that section, we define two metrics, $\textbf{Gain}$ and $\textbf{Correctness}$, to quantify the noise level of pseudo-labels. $\textbf{Gain}$ measures the accuracy gain from the pseudo-label refinement process, and $\textbf{Correctness}$ estimates the effective number of pseudo-labels by multiplying the number of selected pseudo-labels by their accuracy. Through the lens of these two metrics, we empirically find that SEVAL produces pseudo-labels with less noise and provides more effective samples for the SSL process compared with its counterparts.
>
> To further strengthen this observation, we have extended this empirical analysis to more datasets and settings in Section 5.4.

---

> ### Author Response · Authors · 2026-04-03
> **Author Response to Reviewer uicZ for Paper 6142 (2/3)**
>
> > **Q4:** The claim that thresholds should equalise per-class precision relies on strong approximations and is closer to a heuristic than a formal guarantee. The logit-offset formulation largely parallels existing label-shift correction and calibration methods, making the core novelty somewhat incremental.
>
> **A4:** We agree that the proposed methods are not entirely novel in isolation, as they involve sample selection and label shift, which have been addressed by prior work. While many existing methods reduce the bias of majority classes for pseudo-labels, we lack a principled framework to compare them theoretically: we know we should reduce bias from majority classes, but how far we should push the decision boundary remains unclear. We shed new light on this problem by identifying the principled criterion, grounded in statistical theory. We highlight that pseudo-label refinement should not be correlated with the unlabeled data distribution, and that $\textbf{Precision}$, rather than $\textbf{Recall}$, should guide pseudo-label selection. These two conclusions are in fact contrary to common practice in this field, where the unlabeled data distribution is used for calibration and $\textbf{Recall}$ is used for thresholding. We therefore believe this paper makes a timely and important contribution, providing a new standpoint for researchers to rethink pseudo-label adjustment in imbalanced SSL.
>
> > **Q5:** How is the target precision $t$ chosen in practice? Please provide sensitivity analyses across datasets and imbalance ratios, and discuss whether $t$ can be adapted online.
>
> **A5:** The class-wise accuracy $t$ serves as a replacement for the confidence threshold used in existing SSL methods such as FixMatch to filter out low-confidence samples. It is therefore a hyperparameter; we typically choose it in the range of 0.7 to 0.9, and empirically find that classification problems with more classes favor lower values of $t$. In general, the choice is stable across settings. We provide detailed hyperparameter settings in Appendix Section G.2 and have added sensitivity analyses across multiple settings to reflect this. In our pseudo-label quality analysis in Section 5.4, we find that $\textbf{Correctness}$ on $\mathcal{V}$ is correlated with test accuracy, suggesting that it can serve as a practical criterion for online tuning of $t$, where $t$ is updated each epoch to maximize $\textbf{Correctness}$ on $\mathcal{V}$ throughout the curriculum learning process.
>
> We have extended sensitivity analyses on $t$ in Appendix Section F.1 and a discussion of this online tuning strategy in Section 5.4.
>
> > **Q6:** How frequently are $\boldsymbol{\pi}$ and $\boldsymbol{\tau}$ updated during training (curriculum length, EMA coefficients), and what is the computational overhead relative to the base SSL?
>
> **A6:** We typically set the curriculum length $L$ to 500, and use a smaller value when the number of classes is large (e.g., $C = 100$). The optimization of $\boldsymbol{\pi}$ and $\boldsymbol{\tau}$ is lightweight, as it amounts to a constrained optimization problem that can in principle be parallelized with network training, incurring no additional wall-clock time theoretically. The practical overhead stems from the curriculum learning phase conducted prior to standard SSL training, which uses half the training data and thus introduces approximately 50\% additional training time relative to the baseline. We believe this overhead can be further reduced, as the learned curriculum appears stable across different training schedules.
>
> We provide a computational analysis in Section 4.3 and highlight the hyperparameter settings in Section 5.
>
> > **Q7:** Could thresholds be made soft rather than hard 0/1 selection?
>
> **A7:** Hard thresholding is the default in most SSL methods, and we follow this convention. Transitioning to soft thresholds after identifying the confidence threshold is straightforward: samples above the threshold can be upweighted, while those near the threshold can be downweighted. From our perspective, in imbalanced SSL, this is more closely related to long-tail learning or hard-label sampling-based regularization than to pseudo-label adjustment. In this paper, we deliberately avoid extensive long-tail learning or regularization techniques in order to make a fair comparison with existing methods, and instead focus on the core challenges of imbalanced SSL: pseudo-label refinement and threshold adjustment.
>
> We have clarified the focus of SEVAL in Section 5.1, highlighting the effectiveness of SEVAL-PL under this setting and leaving further engineering extensions to future work.

---

> ### Author Response · Authors · 2026-04-03
> **Author Response to Reviewer uicZ for Paper 6142 (3/3)**
>
> > **Q8:** What happens if the test prior is not uniform?
>
> **A8:** Within our framework, adapting to a non-uniform test distribution $P^\mathcal{T}(Y)$ is straightforward: simply reweight the per-class loss accordingly. Experiments on CIFAR-10-LT show that this strategy consistently outperforms baselines when $P^\mathcal{T}(Y)$ deviates from the training distribution. Notably, under an inverse imbalanced test distribution, accuracy improves from 44.6 to 86.0, while gains are more modest when the test distribution matches training (92.4 to 92.5). Existing imbalanced SSL methods lack this adaptability, further supporting the value of a flexible pseudo-label refinement strategy.
>
> We have provided a detailed discussion on non-uniform test distributions in Appendix Section E.

---

### Decision · Action_Editor_XNHr · 2026-04-21

**Recommendation:** Accept with minor revision

**Audience:**

Yes

**Audience Explanation:**

Imbalanced semi-supervised learning is an important task in the field of machine learning.

**Claims And Evidence:**

Yes

**Claims Explanation:**

Two reviewers agree that the claims are fully supported by the evidence.

However, another reviewer is still non-satisfactory, because of the hyperparameter k_c. The authors should always report k_c (number of sample per class in the validation dataset). The authors did not write a response to the above issue.

Thus, the authors should report k_c (number of samples per class in the validation dataset) in the final version.